# Building a stable classifier with the inflated argmax

**Jake A. Soloff**
Department of Statistics
University of Chicago
Chicago, IL 60637
soloff@uchicago.edu

**Rina Foygel Barber**
Department of Statistics
University of Chicago
Chicago, IL 60637
rina@uchicago.edu

**Rebecca Willett**
Departments of Statistics and Computer Science
NSF-Simons National Institute for Theory and Mathematics in Biology
University of Chicago
Chicago, IL 60637
willett@uchicago.edu

## Abstract

We propose a new framework for algorithmic stability in the context of multiclass classification. In practice, classification algorithms often operate by first assigning a continuous score (for instance, an estimated probability) to each possible label, then taking the maximizer—i.e., selecting the class that has the highest score. A drawback of this type of approach is that it is inherently unstable, meaning that it is very sensitive to slight perturbations of the training data, since taking the maximizer is discontinuous. Motivated by this challenge, we propose a pipeline for constructing stable classifiers from data, using bagging (i.e., resampling and averaging) to produce stable continuous scores, and then using a stable relaxation of argmax, which we call the "inflated argmax", to convert these scores to a set of candidate labels. The resulting stability guarantee places no distributional assumptions on the data, does not depend on the number of classes or dimensionality of the covariates, and holds for any base classifier. Using a common benchmark data set, we demonstrate that the inflated argmax provides necessary protection against unstable classifiers, without loss of accuracy.

## 1 Introduction

An algorithm that learns from data is considered to be *stable* if small perturbations of the training data do not lead to large changes in its output. Algorithmic stability is an important consideration in many statistical applications. Within the fairness literature, for instance, stability is one aspect of reliable decision-making systems [FSVC19; HV19]. In interpretable machine learning, it similarly serves as a form of reproducibility [MSKA19; YK20; YB24]. [CMX11] relates stability to robustness, where a machine learning algorithm is robust if two samples with similar feature vectors have similar test error. In the context of generalization bounds, [BE02] and subsequent authors study stability of an algorithm's real-valued output—for instance, values of a regression function. In the setting of a multiclass classification problem, where the data consists of features $X \in \mathcal{X}$ and labels $Y \in [L] = \{1, \ldots, L\}$, the results of this literature can thus be applied to analyzing a classifier's predicted probabilities $\hat{p}_\ell(X) \in [0, 1]$—i.e., our learned estimates of the conditional probabilities $\Pr\{Y = \ell \mid X\}$, for each $\ell \in [L]$. However, as we will see, stability of these predicted probabilities by no means implies stability of the predicted label itself, $\hat{y} = \text{argmax}_{\ell \in [L]} \hat{p}_\ell(x)$—an arbitrarily

38th Conference on Neural Information Processing Systems (NeurIPS 2024).

small perturbation in $\hat{p}_\ell(x)$ might completely change the predicted label $\hat{y}$. The distinction matters: for system trustworthiness, we care about the model's final output, on which users base their decisions.

In this paper, we propose a new framework for algorithmic stability in the context of multiclass classification, to define—and achieve—a meaningful notion of stability when the output of the algorithm consists of predicted labels, rather than predicted probabilities. Our work connects to other approaches to allowing for ambiguous classification, including set-valued classification [Gry93; DDB09; SLW19; CDHL21] and conformal inference [PPVG02; VGS05; Lei14; AB23] (we will discuss related work further, below).

## 1.1 Problem setting

In supervised classification, we take a data set $\mathcal{D} = \big((X_i, Y_i)\big)_{i \in [n]}$ consisting of $n$ observations, and train a classifier that maps from the feature space $\mathcal{X}$ to the set $[L]$ of possible labels.[1] Typically, this map is constructed in two stages. First we run some form of regression to learn a map $\hat{p} : \mathcal{X} \to \Delta_{L-1}$ from features to predicted probabilities, with $\hat{p}_\ell(x)$ denoting our learned estimate of the conditional label probability, $\Pr\{Y = \ell \mid X = x\}$ (here $\Delta_{L-1} = \{w \in \mathbb{R}^L : w_i \geq 0, \sum_i w_i = 1\}$ denotes the probability simplex in $L$ dimensions). We will write $\hat{p} = \mathcal{A}(\mathcal{D})$, where $\mathcal{A}$ denotes the learning algorithm mapping a data set $\mathcal{D}$ (of any size) to the map $\hat{p}$. Then the second step is to convert the predicted probabilities $\hat{p}_\ell(x)$ to a predicted label, which is typically done by taking the argmax, i.e., $\hat{y} = \operatorname{argmax}_\ell \hat{p}_\ell(x)$ (with some mechanism for breaking ties). This two-stage procedure can be represented in the following diagram:

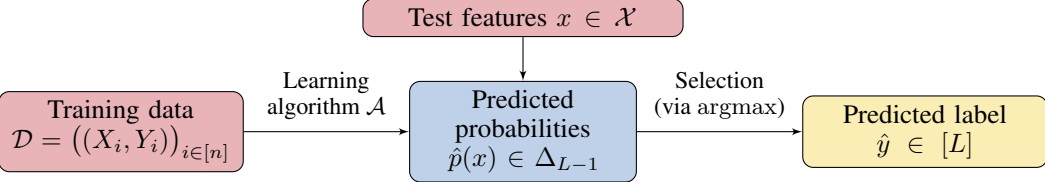

When predictions are ambiguous, meaning two or more classes nearly achieve the maximum predicted probability, the selected label becomes unstable and can change based on seemingly inconsequential perturbations to the training data. In other words, the above workflow is fundamentally incompatible with the goal of algorithmic stability. Consequently, in this paper we instead work with *set-valued classifiers*, which return a set of candidate labels—in practice, this typically leads to a singleton set for examples where we have high confidence in the label, but allows for a larger set in the case of ambiguous examples. While the idea of returning a set of candidate labels is not itself new, we will see that the novelty of our work lies in finding a construction that offers provable stability guarantees.

In this framework, a feature vector $x \in \mathcal{X}$ is now mapped to a set of candidate labels $\hat{S} \subseteq [L]$ (rather than a single label $\hat{y}$), via a selection rule $s$, as illustrated here:

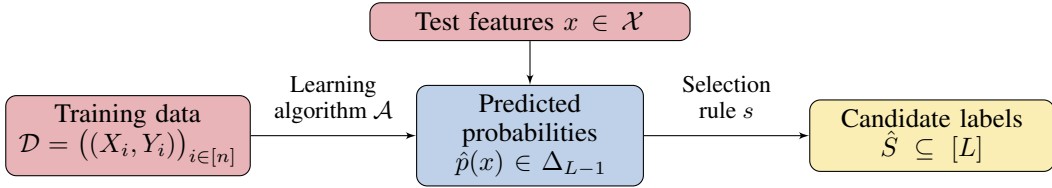

Formally, given a test point $x \in \mathcal{X}$, this two-stage procedure returns $\hat{S} = s(\hat{p}(x)) \subseteq [L]$, where $\hat{p} = \mathcal{A}(\mathcal{D})$ is the output of the regression algorithm $\mathcal{A}$ trained on data $\mathcal{D}$. Here $s : \Delta_{L-1} \to \wp([L])$ denotes a *selection rule*, mapping a vector of estimated probabilities to the set of candidate labels, and $\wp([L])$ denotes the set of subsets of $[L]$. Of course, the earlier setting—where the procedure returns a single label $\hat{y} = \operatorname{argmax}_\ell \hat{p}_\ell(x)$, rather than a subset—can be viewed as a special case by simply taking $s$ to be the argmax operator.

---

[1] We remark that all the results of this paper apply also to the case of countably infinitely many labels, $L = \infty$, in which case we should take $\mathbb{N}$ to be the label space instead of $[L]$.

If we instead allow for $\hat{S}$ to contain multiple candidate labels, a commonly used version of this framework is given by a top-5 procedure (or top-$k$, for any fixed $k$). That is, after running a learning algorithm $\mathcal{A}$ to estimate probabilities $\hat{p}(x)$, the selection rule $s$ then returns the top 5 labels—the labels $\ell_1, \ldots, \ell_5 \in [L]$ corresponding to the highest 5 values of $\hat{p}_\ell(x)$. This approach is more stable than a standard argmax. However, the choice of 5 in this setting is somewhat arbitrary, and the set $\hat{S}$ always contains 5 candidate labels, regardless of the difficulty of the test sample—while intuitively, we might expect that it should be possible to return a smaller $\hat{S}$ for test points $x$ that are easier to classify (and a larger $\hat{S}$ if $x$ is more ambiguous). In contrast, in our work we seek a more principled approach, where we provide provable stability guarantees while also aiming for the set $\hat{S}$ to be as small as possible.

## 1.2 Overview of main results

In this work, we introduce *selection stability*, a new definition of algorithmic stability in the context of classification, which focuses on the stability of the predicted label. We reduce the problem of stabilizing a classifier to separately stabilizing the learning and selection stages, described above. For the selection rule $s$ of the two-stage procedure, we propose the *inflated argmax* operation:

**Definition 1** (Inflated argmax). *For any $w \in \mathbb{R}^L$, define the inflated argmax as*

$$\mathrm{argmax}^\varepsilon(w) := \left\{ j \in [L] : \mathrm{dist}(w, R_j^\varepsilon) < \varepsilon \right\}, \tag{1}$$

*where* $\mathrm{dist}(w, R_j^\varepsilon) = \inf_{v \in R_j^\varepsilon} \|w - v\|$, *and where*

$$R_j^\varepsilon = \left\{ w \in \mathbb{R}^L : w_j \geq \max_{\ell \neq j} w_\ell + \varepsilon/\sqrt{2} \right\}.$$

Our procedure will then return the set $\hat{S} = \mathrm{argmax}^\varepsilon(\hat{p}(x))$ of candidate labels—intuitively, $j \in \mathrm{argmax}^\varepsilon(\hat{p}(x))$ indicates that a small perturbation of the predicted probabilities, $\hat{p}(x)$, would lead to the $j$th label's predicted probability being largest by some margin. In particular, by construction, the inflated argmax will always include any maximal index—that is, if $\hat{p}(x)_j$ is the (possibly non-unique) largest estimated probability, then we must have $j \in \hat{S} = \mathrm{argmax}^\varepsilon(\hat{p}(x))$ (this fact, and other properties of the inflated argmax, will be established formally in Proposition 10 below).

In this work, we derive the stabilizing properties of the inflated argmax, and give an algorithm to compute it efficiently. We prove that combining this operation with bagging at the learning step will provably stabilize any classifier. In particular, our guarantee holds with no assumptions on the data, and no constraints on the dimensionality of the covariates nor on the number of classes.

## 2 Framework: stable classification

In this section, we propose a definition of algorithmic stability in the setting of multiclass classification. To begin, we formally define a *classification algorithm* as a map[2]

$$\mathcal{C} : \cup_{n \geq 0} (\mathcal{X} \times [L])^n \times \mathcal{X} \longrightarrow \wp([L]),$$

which maps a training data set $\mathcal{D}$ of any size $n$, together with a test feature vector $x \in \mathcal{X}$, to a candidate set of labels $\hat{S} = \mathcal{C}(\mathcal{D}, x)$. To relate this notation to our earlier terminology, the two-stage selection procedure described in Section 1.1 corresponds to the classification algorithm

$$\mathcal{C}(\mathcal{D}, x) = s(\hat{p}(x)) \text{ where } \hat{p} = \mathcal{A}(\mathcal{D}).$$

Abusing notation, we will write this as $\mathcal{C} = s \circ \mathcal{A}$, indicating that $\mathcal{C}$ is obtained by applying the selection rule $s$ to the output of the learning algorithm $\mathcal{A}$.

We now present our definition of algorithmic stability for a classification algorithm $\mathcal{C}$. As is common in the algorithmic stability literature, we focus on the stability of the algorithm's output with respect

---

[2]We will assume without comment that $\mathcal{C}$ is measurable. Furthermore, in practice classification algorithms often incorporate randomization (either in the learning stage, such as via stochastic gradient descent, and/or in the selection stage, such as by using a random tie-breaking rule for $\mathrm{argmax}$). All of our definitions and results in this paper can be naturally extended to randomized algorithms—see Appendix B.

to randomly dropping a data point: does the output of $\mathcal{C}$ on a test point $x$ change substantially if we drop a single data point from the training data $\mathcal{D}$? If the algorithm's output were a real-valued prediction $\hat{y} \in \mathbb{R}$, we could assess this by measuring the real-valued change in $\hat{y}$ when a single data point is dropped. For classification, however, we will need to take a different approach:

**Definition 2** (Selection stability). *We say a classification algorithm $\mathcal{C}$ has selection stability $\delta$ at sample size $n$ if, for all datasets $\mathcal{D} \in (\mathcal{X} \times [L])^n$ and all test features $x \in \mathcal{X}$,*

$$\frac{1}{n} \sum_{i=1}^{n} \mathbf{1}\left\{ \hat{S} \cap \hat{S}^{\setminus i} = \varnothing \right\} \leq \delta,$$

*where $\hat{S} = \mathcal{C}(\mathcal{D}, x)$ and where $\hat{S}^{\setminus i} = \mathcal{C}(\mathcal{D}^{\setminus i}, x)$, for each $i \in [n]$.*

Here the notation $\mathcal{D}^{\setminus i}$ denotes the data set $\mathcal{D}$ with $i$th data point removed—that is, for a data set $\mathcal{D} = \left( (X_j, Y_j) \right)_{j \in [n]}$, the leave-one-out data set is given by $\mathcal{D}^{\setminus i} = \left( (X_j, Y_j) \right)_{j \in [n] \setminus i}$.

## 2.1 Interpreting the definition

Intuitively, selection stability controls the probability that our algorithm makes wholly inconsistent claims when dropping a single data point at random from the training set. If we interpret $\hat{S}$ as making the claim that the true label $Y$ is equal to one of the labels in the candidate set $\hat{S}$, and similarly for $\hat{S}^{\setminus i}$, then as soon as there is *even one single value* that lies in both $\hat{S}$ and $\hat{S}^{\setminus i}$, this means that the two claims are not contradictory—even if the sets are large and are mostly nonoverlapping.

At first glance, this condition appears to be too mild, in the sense that we require the two prediction sets only to have *some* way of being compatible, and allows for substantial difference between the sets $\hat{S}$ and $\hat{S}^{\setminus i}$. However, since standard classification algorithms always output a single label, they often cannot be said to be stable even in this basic sense. Thus, we can view this definition as providing a minimal notion of stability that we should require any interpretable method to satisfy.

## 2.2 Connection to classical algorithmic stability

Most prior work on algorithmic stability concerns algorithms with continuous outputs—for example, in our notation above, the algorithm $\mathcal{A}$ that returns estimated probabilities $\hat{p}(x)$. With such algorithms, there are more standard tools at our disposal for quantifying and ensuring stability. In this section, we connect classical algorithmic stability to our notion of selection stability (Definition 2). We first recall the following definition.

**Definition 3** (Tail stability). *[SBW24b] A learning algorithm $\mathcal{A}$ has tail stability $(\varepsilon, \delta)$ at sample size $n$ if, for all datasets $\mathcal{D}$ of size $n$ and all test features $x \in \mathcal{X}$,*

$$\frac{1}{n} \sum_{i=1}^{n} \mathbf{1}\left\{ \|\hat{p}(x) - \hat{p}^{\setminus i}(x)\| \geq \varepsilon \right\} \leq \delta,$$

*where $\hat{p} = \mathcal{A}(\mathcal{D})$ and $\hat{p}^{\setminus i} = \mathcal{A}(\mathcal{D}^{\setminus i})$, and where $\| \cdot \|$ denotes the usual Euclidean norm on $\mathbb{R}^L$.*

This intuitive notion of stability is achieved by many well-known algorithms $\mathcal{A}$, such as nearest-neighbor type methods or methods based on bagging or ensembling (as established by [SBW24a; SBW24b]—see Section 3.1 below for details).

## 2.3 From stability to selection stability

To construct a classification algorithm using the two-stage procedure outlined in Section 1.1, we need to apply a selection rule $s$ to the output of our learning algorithm $\mathcal{A}$. We might expect that choosing a stable $\mathcal{A}$ will lead to selection stability in the resulting classification algorithm—but in fact, this is not the case: even if the learning algorithm $\mathcal{A}$ is itself extremely stable in the sense of Definition 3, the classification rule obtained by applying $\operatorname{argmax}$ to the output of $\mathcal{A}$ can still be extremely unstable. The underlying issue is that $\operatorname{argmax}$ is extremely discontinuous—the perturbation $\|\hat{p}(x) - \hat{p}^{\setminus i}(x)\|$ in the predicted probabilities can be arbitrarily small but still lead to different predicted labels, i.e., $\operatorname{argmax}_\ell \hat{p}_\ell(x) \neq \operatorname{argmax}_\ell \hat{p}_\ell^{\setminus i}(x)$.

Since combining $\mathrm{argmax}$ with a stable learning algorithm $\mathcal{A}$ will not suffice, we instead seek a different selection rule $s$—one that will ensure selection stability (when paired with a stable learning algorithm $\mathcal{A}$). To formalize this aim, we introduce another definition:

**Definition 4** ($\varepsilon$-compatibility). *A selection rule $s : \Delta_{L-1} \to \wp([L])$ is $\varepsilon$-compatible if, for any $v, w \in \Delta_{L-1}$,*

$$\|v - w\| < \varepsilon \implies s(v) \cap s(w) \neq \varnothing.$$

This notion of $\varepsilon$-compatibility allows us to construct classification algorithms with selection stability. Combining the above definitions leads immediately to the following result:

**Proposition 5.** *Let $\mathcal{A}$ be a learning algorithm with tail stability $(\varepsilon, \delta)$ at sample size $n$, and let $s$ be a selection rule satisfying $\varepsilon$-compatibility. Then the classification algorithm $\mathcal{C} = s \circ \mathcal{A}$ has selection stability $\delta$ at sample size $n$.*

Therefore, by pairing a stable learning algorithm $\mathcal{A}$ with a compatible selection rule $s$, we will automatically ensure selection stability of the resulting classification algorithm $\mathcal{C} = s \circ \mathcal{A}$.

Of course, $\varepsilon$-compatibility of the selection rule $s$ might be achieved in a trivial way—for instance, $s$ returns the entire set $[L]$ for any input. As is common in statistical settings (e.g., a tradeoff between Type I error and power, in hypothesis testing problems), our goal is to ensure selection stability while returning an output $\hat{S}$ that is as informative as possible. In particular, later we will consider the specific aim of constructing $\hat{S}$ to be a singleton set as often as possible.

# 3   Methodology: assumption-free stable classification

In this section, we formally define our pipeline for building a stable classification procedure using any base learning algorithm $\mathcal{A}$. At a high level, we leverage Proposition 5 and separately address the learning and selection stages described in Section 2.

1. In Section 3.1, we construct a bagged (i.e., ensembled) version of the base learning algorithm $\mathcal{A}$. The recent work of [SBW24b] ensures that the resulting bagged algorithm has tail stability $(\varepsilon, \delta)$, with $\varepsilon \asymp \frac{1}{\sqrt{n\delta}}$.

2. In Section 3.2, we introduce a new selection rule, the inflated argmax, and establish its $\varepsilon$-compatibility. Combined with the bagged algorithm, then, the resulting classification algorithm will be guaranteed to have selection stability $\delta$ (by Proposition 5).

Before describing these two steps in detail, we first present our main theorem that characterizes the selection stability guarantee of the resulting procedure. Given sample size $n$ for the training data set $\mathcal{D}$, the notation $\widetilde{A}_m$ will denote a bagged version of the base algorithm $\mathcal{A}$, obtained by averaging over subsamples of $\mathcal{D}$ comprised of $m$ data points sampled either with replacement ("bootstrapping", commonly run with $m = n$) or without replacement ("subbagging", commonly run with $m = n/2$)—see below for details.

**Theorem 6.** *Fix any sample size $n$, any bag size $m$, and any inflation parameter $\varepsilon > 0$. For any base learning algorithm $\mathcal{A}$, the classification algorithm $\mathcal{C} = \mathrm{argmax}^\varepsilon \circ \widetilde{A}_m$, obtained by combining the bagged version of $\mathcal{A}$ together with the inflated argmax, satisfies selection stability $\delta$ where*

$$\delta = \varepsilon^{-2} \cdot \frac{1 - 1/L}{n - 1} \cdot \frac{p_{n,m}}{1 - p_{n,m}}, \tag{2}$$

*where $p_{n,m} = 1 - (1 - \frac{1}{n})^m$ for bootstrapping, and $p_{n,m} = \frac{m}{n}$ for subbagging.*

The guarantee in Theorem 6 holds for any base learning algorithm $\mathcal{A}$, and applies regardless of the complexity of the feature space $\mathcal{X}$, and allows the test feature $x \in \mathcal{X}$ to be chosen adversarially. The dependence on the number of classes $L$ is mild—in fact, the tail stability parameter $\delta$ in (2) differs only by a factor of two for $L = 2$ versus $L = \infty$. Of course, the guarantee does depend on the choice of the bag size $m$. In general, a smaller value of $m$ leads to a stronger stability guarantee (since $p_{n,m}$ increases with $m$), but this comes at the expense of accuracy since we are training the base algorithm $\mathcal{A}$ on subsampled data sets $\mathcal{D}^r$ of size $m$ (rather than $n$). For the common choices of $m = n$ for bootstrap or $m = n/2$ for subbagging, we have $\frac{p_{n,m}}{1 - p_{n,m}} = \mathcal{O}(1)$ for each.

### 3.1 Bagging classifier weights

In this section, we formally define the construction of a bagged algorithm to recap the tail stability result that our framework leverages. We consider the two most common variants of this ensembling method: bagging (based on bootstrapping training points) and subbagging (based on subsampling). For any data set $\mathcal{D} \in (\mathcal{X} \times [L])^n$, we can define a subsampled data set of size $m$ as follows: for a sequence $r = (i_1, \ldots, i_m) \in [n]^m$ (which is called a *bag*), we define the corresponding subsampled data set $\mathcal{D}^r = \big((X_i, Y_i)\big)_{i \in r} \in (\mathcal{X} \times [L])^m$. Note that if the bag $r$ contains repeated indices (i.e., $i_k = i_\ell$ for some $k \neq \ell$), then the same data point from the original data set $\mathcal{D}$ will appear multiple times in $\mathcal{D}^r$.

**Definition 7** (Bootstrapping or subbagging a base learning algorithm $\mathcal{A}$)**.** *Given a data set $\mathcal{D} \in (\mathcal{X} \times [L])^n$ and some $x \in \mathcal{X}$, return the output $\widetilde{\mathcal{A}}_m(\mathcal{D})(x) = \mathbb{E}_r[\mathcal{A}(\mathcal{D}^r)(x)]$, where the expected value is taken with respect to a bag $r$ that is sampled as follows:*

- *Bootstrapping (sometimes simply referred to as bagging) [Bre96a; Bre96b] constructs each bag $r$ by sampling $m$ indices $r = (i_1, \ldots, i_m)$ uniformly with replacement from $[n]$.*

- *Subbagging [AEEP02] constructs each bag $r$ by sampling $m \leq n$ indices $r = (i_1, \ldots, i_m)$ uniformly without replacement from $[n]$.*

The following result [SBW24b] ensures tail stability for any bootstrapped or subbagged algorithm:

**Theorem 8** ([SBW24b])**.** *For any base learning algorithm $\mathcal{A}$ returning outputs in $\Delta_{L-1}$, the bagged algorithm $\widetilde{\mathcal{A}}_m$ has tail stability $(\varepsilon, \delta)$ for any $\varepsilon, \delta > 0$ satisfying* (2).

**Computational cost of bagging.** In this section, we have worked with the ideal, derandomized version of bagging for simplicity—that is, we assume that the expected value $\mathbb{E}_r[\mathcal{A}(\mathcal{D}^r)(x)]$ is calculated exactly with respect to the distribution of $r$. In practice, of course, this is computationally infeasible (since for bootstrapping, say, there are $n^m$ possible bags $r$), and so we typically resort to Monte Carlo sampling to approximate this expected value, defining $\widetilde{\mathcal{A}}_m(\mathcal{D})(x) = \frac{1}{B}\sum_{b=1}^B \mathcal{A}(\mathcal{D}^{r_b})(x)$, for $B$ i.i.d. bags $r_1, \ldots, r_B$ sampled via bootstrapping or subbagging. See Appendix C for extensions of our main result, Theorem 6, to the finite-$B$ version of the method.

### 3.2 The inflated argmax

We now return to the inflated argmax operator, as defined in Definition 1. This operator, in place of the standard argmax, allows for stability in classification.

While $\mathrm{argmax}^\varepsilon$ is defined as an operator on $\mathbb{R}^L$, in the context of classification algorithms, we only apply it to vectors $w$ lying in the simplex $\Delta_{L-1}$ (in particular, in the context of a two-stage classification procedure as in Section 1.1, we will apply the inflated argmax to the vector $w = \hat{p}(x)$ of estimated probabilities). In Figure 1, we visualize the inflated argmax applied to the simplex in a setting with three possible labels, $L = 3$. Each different shaded area corresponds to the region of the simplex $\Delta_{L-1}$ where $\mathrm{argmax}^\varepsilon(w)$ returns a particular subset.

#### 3.2.1 Inflated argmax leads to selection stability

Unlike the standard argmax, the inflated argmax allows us to achieve stability in the context of classification. This is due to the following theorem, which shows that the inflated argmax is $\varepsilon$-compatible, as introduced in Definition 4. (The proofs of this result and all properties of the inflated argmax are deferred to Appendix A.)

**Theorem 9.** *For any $\varepsilon > 0$ and any $v, w \in \mathbb{R}^L$, if $\|v - w\| < \varepsilon$ then $\mathrm{argmax}^\varepsilon(v) \cap \mathrm{argmax}^\varepsilon(w) \neq \varnothing$. In particular, viewing $\mathrm{argmax}^\varepsilon$ as a map on the simplex $\Delta_{L-1}$, $\mathrm{argmax}^\varepsilon$ is $\varepsilon$-compatible.*

To gain a more concrete understanding of this theorem, we can look again at Figure 1 to examine the case $L = 3$. Theorem 9 ensures that any two regions corresponding to disjoint subsets—e.g., the top corner region corresonding to $\{2\}$, and the bottom center region corresponding to $\{1, 3\}$—are distance at least $\varepsilon$ apart. In particular, the region in the center, corresponding to vectors $w$ that map to the full set $\{1, 2, 3\}$, is a curved triangle of constant width $\varepsilon$, also known as a *Reuleaux triangle* [Reu76; Wei05].

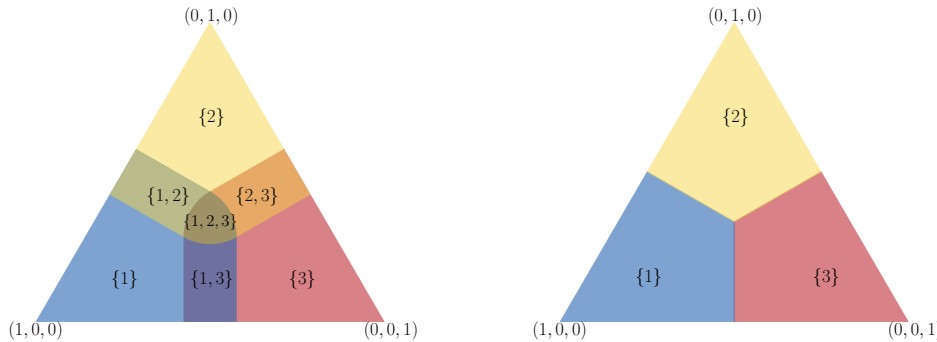

Figure 1: The left plot illustrates the inflated argmax (1) over the simplex $\Delta_{L-1}$ when $L = 3$. The numbers in brackets correspond to the output of the inflated argmax, $\mathrm{argmax}^\varepsilon(w)$, for various points $w$ in the simplex. The right plot shows the same but for the standard argmax, which corresponds to the limit of $\mathrm{argmax}^\varepsilon(w)$ as $\varepsilon \to 0$.

**Proof of Theorem 6.**    With the above results in place, we can now see that the inflated argmax allows us to achieve selection stability, when combined with a bagged algorithm. In particular, combining Theorem 8, Theorem 9, and Proposition 5 immediately implies our main result, Theorem 6.

### 3.2.2   Additional properties of the inflated argmax

The following result establishes some natural properties obeyed by the inflated argmax, and also compares to the standard argmax.

**Proposition 10** (Basic properties of the inflated argmax)**.** *Fix any $\varepsilon > 0$. The inflated argmax operator satisfies the following properties:*

1. *(Including the argmax.)  For any $w \in \mathbb{R}^L$, $\mathrm{argmax}(w) \subseteq \mathrm{argmax}^\varepsilon(w)$.[3]  Moreover, $\cap_{\varepsilon>0}\, \mathrm{argmax}^\varepsilon(w) = \mathrm{argmax}(w)$.*

2. *(Monotonicity in $\varepsilon$.)  For any $w \in \mathbb{R}^L$ and any $\varepsilon < \varepsilon'$, $\mathrm{argmax}^\varepsilon(w) \subseteq \mathrm{argmax}^{\varepsilon'}(w)$.*

3. *(Monotonicity in $w$.)  For any $w \in \mathbb{R}^L$, if $w_j \leq w_k$, then $j \in \mathrm{argmax}^\varepsilon(w) \Rightarrow k \in \mathrm{argmax}^\varepsilon(w)$.*

4. *(Permutation invariance.)  For any $v, w \in \mathbb{R}^L$, if $v = (w_{\sigma(1)}, \ldots, w_{\sigma(L)})$ for some permutation $\sigma$ on $[L]$, then $j \in \mathrm{argmax}^\varepsilon(v) \Leftrightarrow \sigma(j) \in \mathrm{argmax}^\varepsilon(w)$.*

Next, while we have established that inflated argmax offers favorable stability properties, we have not yet asked whether it can be efficiently computed—in particular, it is not immediately clear how to verify the condition $\mathrm{dist}(w, R_j^\varepsilon) < \varepsilon$ in (1), in order to determine whether $j \in \mathrm{argmax}^\varepsilon(w)$. The following result offers an efficient algorithm for computing the inflated argmax set.

**Proposition 11** (Computing the inflated argmax)**.** *Fix any $w \in \mathbb{R}^L$ and $\varepsilon > 0$. Let $w_{[1]} \geq \cdots \geq w_{[L]}$ denote the order statistics of $w$, and define*

$$\hat{k}(w) = \max\left\{ k \in [L] : \left(\sum_{\ell=1}^{k}(w_{[\ell]} - w_{[k]})\right)^2 + \sum_{\ell=1}^{k}(w_{[\ell]} - w_{[k]})^2 \leq \varepsilon^2 \right\}.$$

*Then*

$$\mathrm{argmax}^\varepsilon(w) = \left\{ j \in [L] : w_j > \frac{\varepsilon}{\sqrt{2}} + \hat{A}_1(w) - \sqrt{\hat{k}(w) + 1}\sqrt{\frac{\varepsilon^2}{\hat{k}(w)} + (\hat{A}_1(w))^2 - \hat{A}_2(w)} \right\},$$

*where $\hat{A}_1(w) = \frac{w_{[1]} + \cdots + w_{[\hat{k}(w)]}}{\hat{k}(w)}$, and $\hat{A}_2(w) = \frac{w_{[1]}^2 + \cdots + w_{[\hat{k}(w)]}^2}{\hat{k}(w)}$.*

---

[3]In this result, $\mathrm{argmax}(w)$ should be interpreted as the set of *all* maximizing entries of $w$—i.e., in the case of ties, $\mathrm{argmax}(w)$ may not be a singleton set.

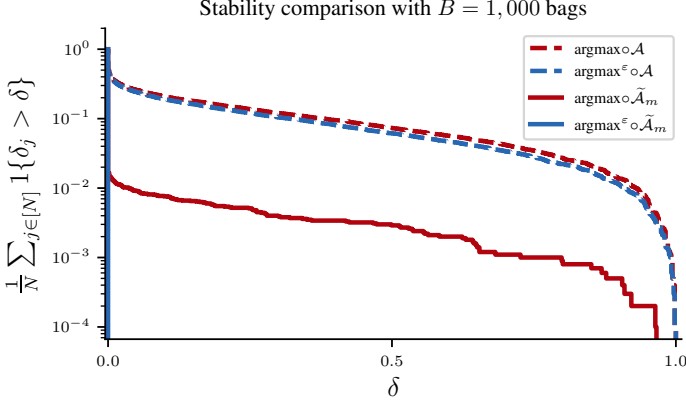

Figure 2: Results on the Fashion MNIST data set. The figure shows the instability $\delta_j$ (defined in (4)) over all test points $j = 1, \ldots, N$. The curves display the fraction of $\delta_j$'s that exceed $\delta$, for each value $\delta \in [0, 1]$. The vertical axis is on a log scale. See Section 4 for details.

Finally, thus far we have established that $\mathrm{argmax}^\varepsilon$ enables us to achieve stable classification with a computationally efficient selection rule—but we do not yet know whether $\mathrm{argmax}^\varepsilon$ is optimal, or whether some other choice of $s$ might lead to smaller output sets $\hat{S}$ (while still offering assumption-free stability). For instance, we might consider a fixed-margin rule,

$$s^\varepsilon_{\mathrm{margin}}(w) = \{j : w_j > \max_\ell w_\ell - \varepsilon/\sqrt{2}\}, \tag{3}$$

for which $\varepsilon$-compatibility also holds—might this simpler rule be better than the inflated argmax? Our next result establishes that—under some natural constraints—$\mathrm{argmax}^\varepsilon$ is in fact the optimal choice of selection rule, in the sense of returning a singleton set (i.e., $|\hat{S}| = 1$) as often as possible.

**Proposition 12** (Optimality of the inflated argmax). *Let $s : \Delta_{L-1} \to \wp([L])$ be any selection rule. Suppose $s$ is $\varepsilon$-compatible (Definition 4), permutation invariant (in the sense of Proposition 10), and contains the argmax. Then for any $w \in \Delta_{L-1}$ and any $j \in [L]$,*

$$s(w) = \{j\} \implies \mathrm{argmax}^\varepsilon(w) = \{j\}.$$

In other words, for any selection rule $s$ satisfying the assumptions of the proposition (which includes the fixed-margin rule, $s^\varepsilon_{\mathrm{margin}}$), if $s(w)$ is a singleton set then so is $\mathrm{argmax}^\varepsilon(w)$. (See also Appendix D for a closer comparison between the inflated argmax and the fixed-margin selection rule given in (3).)

## 4 Experiments

In this section, we evaluate our proposed pipeline, combining subbagging with the inflated argmax, with deep learning models and on a common benchmark data set.[4]

**Data and models.** We use Fashion-MNIST [XRV17], which consists of $n = 60,000$ training pairs $(X_i, Y_i)$, $N = 10,000$ test pairs $(\tilde{X}_j, \tilde{Y}_j)$, and $L = 10$ classes. For each data point $(X, Y)$, $X$ is a $28 \times 28$ grayscale image that pictures a clothing item, and $Y \in [L]$ indicates the type of item, e.g., a dress, a coat, etc. The base model we use is a variant of LeNet-5, implemented in PyTorch [PGML19] tutorials as GarmentClassifier(). The base algorithm $\mathcal{A}$ trains this classifier using 5 epochs of stochastic gradient descent.

**Methods and evaluation.** We compare four methods:

1. The argmax of the base learning algorithm $\mathcal{A}$.

---

[4]Code to fully reproduce the experiment is available at `https://github.com/jake-soloff/stable-argmax-experiments`. Training all of the models for this experiment took a total of four hours on 10 CPUs running in parallel on a single computing cluster.

| Selection rule | Algo. | $\beta_{\text{correct-single}}$ ↗ | $\beta_{\text{set-size}}$ ↘ | $\beta_{\text{max-instability}}$ ↘ |
|---|---|---|---|---|
| argmax | $\mathcal{A}$ | 0.879 (0.003) | 1.000 (0.000) | 1.000 |
| $\text{argmax}^\varepsilon$ | $\mathcal{A}$ | 0.873 (0.003) | 1.015 (0.001) | 1.000 |
| argmax | $\widetilde{\mathcal{A}}_m$ | 0.893 (0.003) | 1.000 (0.000) | 0.966 |
| $\text{argmax}^\varepsilon$ | $\widetilde{\mathcal{A}}_m$ | 0.886 (0.003) | 1.018 (0.001) | 0.000 |

Table 1: Results on the Fashion MNIST data set. We display the frequency of returning the correct label as a singleton, $\beta_{\text{correct-single}}$ and the average size, $\beta_{\text{set-size}}$, both of which are defined in (6). Standard errors for these averages are shown in parentheses. The final column shows the worst-case instability over the test set, $\beta_{\text{max-instability}}$, defined in (5). For each metric, the symbol ↗ indicates that higher values are desirable, while ↘ indicates that lower values are desirable.

2. The $\varepsilon$-inflated argmax of the base learning algorithm $\mathcal{A}$ with tolerance $\varepsilon = .05$.

3. The argmax of the subbagged algorithm $\widetilde{\mathcal{A}}_m$, with $B = 1,000$ bags of size $m = n/2$.

4. The $\varepsilon$-inflated argmax of the subbagged algorithm $\widetilde{\mathcal{A}}_m$, with $B = 1,000$ bags of size $m = n/2$ and tolerance $\varepsilon = .05$.

We evaluate each method based on several metrics. First, to assess selection stability, for each test point $j = 1, \ldots, N$ we compute its instability

$$\delta_j := \frac{1}{500} \sum_{k=1}^{500} \mathbf{1}\left\{ s(\hat{p}(\tilde{X}_j)) \cap s(\hat{p}^{\backslash i_k}(\tilde{X}_j)) = \varnothing \right\}, \tag{4}$$

where $i_1, \ldots, i_{500}$ are sampled uniformly without replacement from $[n]$ (i.e., we are sampling 500 leave-one-out models $\hat{p}^{\backslash i}$ to compare to the model $\hat{p}$ trained on the full training set). Since our theory concerns worst-case instability over all possible test points, we evaluate the maximum instability

$$\beta_{\text{max-instability}} := \max_{j \in [N]} \delta_j. \tag{5}$$

Second, to evaluate how accurate each method is, we compute how often the method returns the correct label as a singleton $\beta_{\text{correct-single}}$ and set size $\beta_{\text{set-size}}$ (the number of labels in the candidate set):

$$\beta_{\text{correct-single}} := \frac{1}{N} \sum_{j=1}^{N} \mathbf{1}\left\{ s(\hat{p}(\tilde{X}_j)) = \{\widetilde{Y}_j\} \right\}, \quad \beta_{\text{set-size}} := \frac{1}{N} \sum_{j=1}^{N} \left| s(\hat{p}(\tilde{X}_j)) \right|. \tag{6}$$

Ideally we would want a method to return the correct singleton as frequently as possible (a large value of $\beta_{\text{correct-single}} \in [0, 1]$ that is close to 1) and small set size (a value of $\beta_{\text{set-size}} \geq 1$ that is close to 1).

**Results.** In Figure 2, we present results for the instability of each method, plotting the survival function of the instability for all test points $(\delta_j)_{j \in [N]}$. The standard argmax applied to the base algorithm, $\text{argmax} \circ \mathcal{A}$, has the longest tail, meaning $\delta_j$ is large for many test points $j$. The inflated argmax applied to the base algorithm, $\text{argmax}^\varepsilon \circ \mathcal{A}$, offers only a very small improvement on the stability. By contrast, applying the standard argmax to the subbagged algorithm, $\text{argmax} \circ \widetilde{\mathcal{A}}_m$, offers a much more substantial improvement, since the red dashed curve is much smaller than both of the solid curves. Still, some of the largest $\delta_j$ for this method are near 1, meaning for these test points the returned set is sensitive to dropping a single training instance. Combining the inflated argmax with subbagging, $\text{argmax}^\varepsilon \circ \widetilde{\mathcal{A}}_m$, offers a dramatic improvement: in this case $\delta_j = 0$ for every $j = 1, \ldots, N$.

In Table 1, we present the average measures, $\beta_{\text{correct-single}}$ and $\beta_{\text{set-size}}$ and the worst-case measure $\beta_{\text{max-instability}}$. For both the base algorithm and the subbagged algorithm, applying the inflated argmax incurs a small cost both in terms of the frequency of returning the correct label as a singleton and the average size. The inflated argmax increases set size only very slightly, but when combined with subbagging, we improve upon the $\beta_{\text{correct-single}}$ of the original method and achieve an extremely high level of stability ($\beta_{\text{max-instability}} = 0$), while returning a singleton set in the vast majority of cases.

We extend our experiment in Appendix E. In particular, we compare the inflated argmax to some existing methods for set-valued classification, including simple thresholding and top-$K$ rules as well as more involved methods, all of which are Bayes rules for various utility functions [MWDH21]. We also evaluate each method using some additional metrics from set-valued classification, including utility-discounted predictive accuracy [ZCM12].

## 5 Discussion

**Related work.** There is an enormous literature on set-valued classification, so we only reference some of the most relevant works. Much prior work has considered the possibility of a more relaxed argmax to allow for uncertainty [Bri90; BV16; MA16; BMN20; ZLYW23]. The more recent papers in this line of work have focused on producing a sparse set of weights, but none of these works offer a formal stability guarantee for the support of the weights. Our work is the first to propose a version of the argmax that can provably control a notion of stability for the classification setting.

Our definition of selection stability relies on a notion of consistency between sets—two sets of candidate labels are consistent if they have at least one common element. This is similar in flavor to conformal classification [Lei14; SLW19], where a set of candidate values is valid if it contains the correct label; this method does not rely on any distributional assumptions, and consequently has been applied to complex and high-dimensional applications such as generative language models [QFSY23]. These frameworks share the motivation of avoiding 'overconfidence in the assignment of a definite label' [HPW18].

**Overview, limitations and open questions.** We prove a guarantee on the selection stability based on our methodology combining bagging with the inflated argmax. Theorem 6 does not place any assumptions on the learning algorithm nor on the data, including any distributional assumptions. In fact, the training data and test point may be chosen adversarially based on the algorithm, and the output will still be stable. Moreover, we do not assume that the sample size is large: the guarantee holds for any fixed training set size $n$ and improves as $n$ increases. Furthermore, the inflated argmax selection rule ensures that the returned sets of candidate labels are as small as possible (i.e., that there is as little ambiguity as possible about the predicted label for any given test point $x$).

While our theorem does not require assumptions, our method does require bagging the base learning algorithm. The main limitation of our work is that bagging is computationally intensive. However, training different bags is easily parallelizable, which is what allowed us to easily train a convolutional neural network on $B = 1,000$ total subsets of the training data. Moreover, while the original definition of bagging uses the conventional bootstrap, where each bag contains as many samples as the original data set, i.e. $m = n$, in our framework we allow for arbitrary bag size $m$, which could be much smaller than the sample size $n$. Massively subsampling the data ($m \ll n$) can actually help scale learning algorithms to large data sets [KTSJ14]. Moreover, bagging with $m = n$ can be expensive, but there are still many areas of machine learning where it is used, notably in Random Forests. Finally, our experiments also show that a modest number of bags ($B = 1,000$) is all that we really need to start seeing major gains in selection stability.

We leave open several avenues for future work. Practitioners may be interested in other forms of discrete stability, which are more strict than requiring one common element between $\hat{S}$ and $\hat{S}^{\setminus i}$. For instance, one popular way to measure set similarity is the Jaccard index, given by $\frac{|\hat{S} \cap \hat{S}^{\setminus i}|}{|\hat{S} \cup \hat{S}^{\setminus i}|}$. Another open problem is to extend the stability framework and methods studied here to more structured discrete outputs, such as in clustering and variable selection, where nontrivial metrics on the output are more common.

## Acknowledgements

We gratefully acknowledge the support of the National Science Foundation via grant DMS-2023109 DMS-2023109. R.F.B. and J.A.S. gratefully acknowledge the support of the Office of Naval Research via grant N00014-20-1-2337. R.M.W. gratefully acknowledges the support of the NSF-Simons National Institute for Theory and Mathematics in Biology (NSF DMS-2235451, Simons Foundation MP-TMPS-00005320), and AFOSR FA9550-18-1-0166. The authors thank Melissa Adrian for help with running the experiments.

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

# A  Proofs of properties of the inflated argmax

In this section, we prove all the results of Section 3.2, establishing the various properties of the inflated argmax. In fact, all the proofs will rely on the following theorem, which gives an alternative characterization of the set $\mathrm{argmax}^\varepsilon(w)$.

**Theorem 13.** *Fix $\varepsilon > 0$. Define the map $c_\varepsilon : \mathbb{R}^L \to \mathbb{R}$ as*

$$c_\varepsilon(w) \text{ is the unique solution to } \left( \sum_{j \in [L]} (w_j - c)_+ \right)^2 + \sum_{j \in [L]} (w_j - c)_+^2 = \varepsilon^2,$$

*and define the map $t_\varepsilon : \mathbb{R}^L \to \mathbb{R}$ as*

$$t_\varepsilon(w) = c_\varepsilon(w) + \varepsilon/\sqrt{2} - \sum_{j \in [L]} (w_j - c_\varepsilon(w))_+.$$

*Then for any $w \in \mathbb{R}^L$, it holds that*

$$\mathrm{argmax}^\varepsilon(w) = \{j \in [L] : w_j > t_\varepsilon(w)\}.$$

*Moreover, $t_\varepsilon(w) \leq c_\varepsilon(w)$ for all $w$ and all $\varepsilon$, and $t_\varepsilon(w)$ and $c_\varepsilon(w)$ are both nondecreasing functions of $\varepsilon$, and both nondecreasing functions of $w$ (i.e., if $w \leq w'$ holds coordinatewise, then $c_\varepsilon(w) \leq c_\varepsilon(w')$ and $t_\varepsilon(w) \leq t_\varepsilon(w')$).*

With this key result in place, we are now ready to turn to the proofs of the results stated in Section 3.2. Throughout all the proofs below, the functions $c_\varepsilon(w)$ and $t_\varepsilon(w)$ are defined as in the statement of Theorem 13.

## A.1  Proof of Theorem 9

Let $w, w' \in \mathbb{R}^L$ be any vectors with $\mathrm{argmax}^\varepsilon(w) \cap \mathrm{argmax}^\varepsilon(w') = \varnothing$. Define

$$B = \{j \in [L] : w_j > c_\varepsilon(w)\}, \quad B' = \{j \in [L] : w'_j > c_\varepsilon(w')\},$$

and note that we must have $B \subseteq \mathrm{argmax}^\varepsilon(w)$ since for $j \notin \mathrm{argmax}^\varepsilon(w)$, $w_j \leq t_\varepsilon(w) \leq c_\varepsilon(w)$ by Theorem 13. Similarly, $B' \subseteq \mathrm{argmax}^\varepsilon(w')$. We also must have $B$ (and similarly $B'$) nonempty by definition of $c_\varepsilon(w)$.

For convenience, define $c = c_\varepsilon(w)$, $c' = c_\varepsilon(w')$, $t = t_\varepsilon(w)$, $t' = t_\varepsilon(w')$. We then have

$$\|w - w'\|^2 \geq \|w_B - w'_B\|^2 + \|w'_{B'} - w_{B'}\|^2$$
$$= \|(w_B - c\mathbf{1}_B) + (t'\mathbf{1}_B - w'_B) + (c - t')\mathbf{1}_B\|^2 + \|(w'_{B'} - c'\mathbf{1}_{B'}) + (t\mathbf{1}_{B'} - w_{B'}) + (c' - t)\mathbf{1}_{B'}\|^2, \tag{7}$$

where the first step holds since $B \cap B' = \varnothing$ due to $\mathrm{argmax}^\varepsilon(w) \cap \mathrm{argmax}^\varepsilon(w') = \varnothing$. Next we will need a technical lemma.

**Lemma 14.** *Let $k, k' \geq 1$ be integers and let $\varepsilon > 0$. Then for any $x, y \in \mathbb{R}^k_{\geq 0}$, $x', y' \in \mathbb{R}^{k'}_{\geq 0}$, and $r, r' \in \mathbb{R}$, if*

$$(x^\top \mathbf{1}_k)^2 + \|x\|^2 = (x'^\top \mathbf{1}_{k'})^2 + \|x'\|^2 = \varepsilon^2 \tag{8}$$

*and*

$$x^\top \mathbf{1}_k + x'^\top \mathbf{1}_{k'} = r + r' + \varepsilon\sqrt{2}, \tag{9}$$

*then*

$$\|x + y + r\mathbf{1}_k\|^2 + \|x' + y' + r'\mathbf{1}_{k'}\|^2 \geq \varepsilon^2.$$

To apply this lemma, first we let $k = |B| \geq 1$ and $k' = |B'| \geq 1$, and define

$$x = w_B - c\mathbf{1}_B, \ y = t'\mathbf{1}_B - w'_B, \ x' = w'_{B'} - c'\mathbf{1}_{B'}, \ y' = t\mathbf{1}_{B'} - w_{B'},$$

and note that all these vectors have nonnegative entries (specifically, $x$ and $x'$ are nonnegative by definition of $B$ and $B'$, while $y$ and $y'$ are nonnegative by Theorem 13, using the fact that

$B \cap s(w') = B' \cap s(w) = \varnothing$). Then the condition (8) is satisfied by definition of $c = c_\varepsilon(w)$ and $c' = c_\varepsilon(w')$. Define also

$$r = c - t', \ r' = c' - t.$$

and note that

$$r + r' = (c - t) + (c' - t') = \sum_{j \in [L]} (w_j - c)_+ - \varepsilon/\sqrt{2} + \sum_{j \in [L]} (w'_j - c')_+ - \varepsilon/\sqrt{2}$$

$$= x^\top \mathbf{1}_k + {x'}^\top \mathbf{1}_{k'} - \varepsilon\sqrt{2},$$

where the second step holds by definition of $t = t_\varepsilon(w)$ and $t' = t_\varepsilon(w')$. Therefore, (9) is also satisfied. Returning to (7), Lemma 14 then implies that $\|w - w'\|^2 \geq \varepsilon^2$, which completes the proof of the theorem.

## A.2 Proof of Proposition 10

First we verify that $\mathrm{argmax}(w) \subseteq \mathrm{argmax}^\varepsilon(w)$. Let $j \in \mathrm{argmax}(w)$, i.e., $w_j = \max_\ell w_\ell$. Let $v = w + \mathbf{e}_j \cdot \varepsilon/\sqrt{2}$, where $\mathbf{e}_j = (0, \ldots, 0, 1, 0, \ldots, 0)$ is the $j$th canonical basis vector. Then $v \in R_j^\varepsilon$, and $\|w - v\| = \varepsilon/\sqrt{2} < \varepsilon$, so we have $\mathrm{dist}(w, R_j^\varepsilon) < \varepsilon$ and therefore $j \in \mathrm{argmax}^\varepsilon(w)$. We also need to verify that $\cap_{\varepsilon > 0} \mathrm{argmax}^\varepsilon(w) = \mathrm{argmax}(w)$, i.e., for $j \notin \mathrm{argmax}(w)$, there is some $\varepsilon > 0$ with $j \notin \mathrm{argmax}^\varepsilon(w)$. Let $k \in [L]$ be an index with $w_k > w_j$, and let $\varepsilon = \sqrt{2}(w_k - w_j)$. Then for any $v \in R_j^\varepsilon$, we have

$$\|w - v\|^2 \geq (w_j - v_j)^2 + (w_k - v_k)^2 = (w_k - \varepsilon/\sqrt{2} - v_j)^2 + (w_k - v_k)^2$$

$$\geq \inf_{t \in \mathbb{R}} \left\{ \left( t - (\varepsilon/\sqrt{2} + (v_j - v_k)) \right)^2 + t^2 \right\} = \frac{1}{2}(\varepsilon/\sqrt{2} + (v_j - v_k))^2 \geq \varepsilon^2,$$

where the last step holds since $v_j - v_k \geq \varepsilon/\sqrt{2}$. This means that $\mathrm{dist}(w, R_j^\varepsilon) \geq \varepsilon$ and so $j \notin \mathrm{argmax}^\varepsilon(w)$ at this value of $\varepsilon > 0$.

Next we check monotonicity in $\varepsilon$. Fix $\varepsilon < \varepsilon'$. Then for any $j \in [L]$,

$$j \in \mathrm{argmax}^\varepsilon(\varepsilon) \iff w_j > t_\varepsilon(w) \implies w_j > t_{\varepsilon'}(w) \iff j \in \mathrm{argmax}^{\varepsilon'}(w),$$

where each step holds by Theorem 13.

Next we verify monotonicity in $w$. If $w_j \leq w_k$, then by Theorem 13

$$j \in \mathrm{argmax}^\varepsilon(w) \iff w_j > t_\varepsilon(w) \implies w_k > t_\varepsilon(w) \iff k \in \mathrm{argmax}^\varepsilon(w).$$

Finally we check permutation invariance. Suppose $v = (w_{\sigma(1)}, \ldots, w_{\sigma(L)})$ is a permutation of $w$. Then by construction, we have $c_\varepsilon(v) = c_\varepsilon(w)$ and $t_\varepsilon(v) = t_\varepsilon(w)$. In particular,

$$j \in \mathrm{argmax}^\varepsilon(v) \iff v_j > t_\varepsilon(v) \iff w_{\sigma(j)} > t_\varepsilon(w) \iff \sigma(j) \in \mathrm{argmax}^\varepsilon(w).$$

## A.3 Proof of Proposition 11

Define a function $f_w$ as

$$f_w(c) = \left( \sum_{j \in [L]} (w_j - c)_+ \right)^2 + \sum_{j \in [L]} (w_j - c)_+^2. \tag{10}$$

Then $\hat{k}(w)$ can equivalently be defined as

$$\hat{k}(w) = \max\{k \in [L] : f_w(w_{[k]}) \leq \varepsilon^2\}.$$

(Note that this maximum is well defined, since $f_w(w_{[1]}) = 0$ and so the set is nonempty.) Since $c \mapsto f_w(c)$ is strictly decreasing over $c \leq w_{[1]}$, we see that the solution $c_\varepsilon(w)$ to the equation

$f_w(c) = \varepsilon^2$ must satisfy $c_\varepsilon(w) \leq w_{[\hat{k}(w)]}$, and (if $\hat{k}(w) < L$) also $c_\varepsilon(w) > w_{[\hat{k}(w)+1]}$. In particular, this implies

$$\varepsilon^2 = \left(\sum_{j \in [L]} (w_j - c_\varepsilon(w))_+\right)^2 + \sum_{j \in [L]} (w_j - c_\varepsilon(w))_+^2$$

$$= \left(\sum_{j=1}^{\hat{k}(w)} (w_j - c_\varepsilon(w))\right)^2 + \sum_{j=1}^{\hat{k}(w)} (w_j - c_\varepsilon(w))^2$$

$$= \left(\hat{k}(w) \cdot (\hat{A}_1(w) - c_\varepsilon(w))\right)^2 + \left(\hat{k}(w)\hat{A}_2(w) - 2\hat{k}(w)\hat{A}_1(w)c_\varepsilon(w) + \hat{k}(w)c_\varepsilon(w)^2\right)$$

$$= \hat{k}(w)(\hat{k}(w) + 1)\left(c_\varepsilon(w)^2 - 2c_\varepsilon(w)\hat{A}_1(w) + \frac{\hat{k}(w)(\hat{A}_1(w))^2 + \hat{A}_2(w)}{\hat{k}(w) + 1}\right).$$

This is a quadratic function of $c_\varepsilon(w)$, and is solved by

$$c_\varepsilon(w) = \hat{A}_1 - \sqrt{\frac{(\hat{A}_1(w))^2 - \hat{A}_2(w) + \frac{\varepsilon^2}{\hat{k}(w)}}{\hat{k}(w) + 1}}.$$

We also have

$$t_\varepsilon(w) = c_\varepsilon(w) + \varepsilon/\sqrt{2} - \sum_{j \in [L]} (w_j - c_\varepsilon(w))_+$$

$$= c_\varepsilon(w) + \varepsilon/\sqrt{2} - \sum_{j=1}^{\hat{k}(w)} (w_{[j]} - c_\varepsilon(w))$$

$$= (\hat{k}(w) + 1)c_\varepsilon(w) + \frac{\varepsilon}{\sqrt{2}} - \hat{k}(w)\hat{A}_1(w).$$

Plugging in our above expression for $c_\varepsilon(w)$, then,

$$t_\varepsilon(w) = \frac{\varepsilon}{\sqrt{2}} + \hat{A}_1(w) - \sqrt{\hat{k}(w) + 1}\sqrt{(\hat{A}_1(w))^2 - \hat{A}_2(w) + \frac{\varepsilon^2}{\hat{k}(w)}}.$$

Since $\text{argmax}^\varepsilon(w) = \{j : w_j > t_\varepsilon(w)\}$ by Theorem 13, this completes the proof.

### A.4 Proof of Proposition 12

First we will need a lemma:

**Lemma 15.** *Fix $w \in \mathbb{R}^L$. Then $w \in R_j^\varepsilon$ if and only if $\text{argmax}^\varepsilon(w) = \{j\}$.*

For intuition on this result, we can look back at Figure 1—for instance, the blue region marked by $\{1\}$ illustrates the set of vectors $w \in \Delta_{L-1}$ such that $\text{argmax}^\varepsilon(w) = \{1\}$, and according to this lemma, this is equal to the region $R_1^\varepsilon$.

Using this lemma, we now need to show that, for any $\varepsilon$-compatible and permutation invariant selection rule $s$, if $s(w) = \{j\}$ for some $w \in \Delta_{L-1}$ and $j \in [L]$, then $w \in R_j^\varepsilon$ for some $j$. First, we must have $j = \text{argmax}(w)$, since $s$ is assumed to contain the argmax. Fix any $k \neq j$, and define a vector $v \in \Delta_{L-1}$ by permuting $w_j$ and $w_k$:

$$v_\ell = \begin{cases} w_k, & \ell = j, \\ w_j, & \ell = k, \\ w_\ell, & \ell \neq j, k. \end{cases}$$

Then by permutation invariance we have $s(v) = \{k\}$. Since $s$ is $\varepsilon$-compatible, we therefore have

$$\varepsilon \leq \|w - v\| = \sqrt{(w_j - v_j)^2 + (w_k - v_k)^2 + \sum_{\ell \neq j,k} (w_\ell - v_\ell)^2} = \sqrt{2(w_j - w_k)^2}.$$

Therefore we have $w_j - w_k \geq \varepsilon/\sqrt{2}$ for all $k \neq j$, which proves $w \in R_j^\varepsilon$ and therefore $\text{argmax}^\varepsilon(w) = \{j\}$.

## A.5 Proof of Theorem 13

**Step 1: verifying that $c_\varepsilon(w)$ is well defined.** First we check that the solution $c_\varepsilon(w)$ exists and is unique. Fix $w$ and define the function $f_w$ as in (10). Note that $f_w(c) \equiv 0$ for $c \geq \max_i w_i$, and $c \mapsto f_w(c)$ is strictly decreasing over $c \leq \max_i w_i$, with $\lim_{c \to -\infty} f_w(c) = \infty$; therefore, a unique solution $c_\varepsilon(w)$ to the equation $f_w(c) = \varepsilon^2$ must exist.

**Step 2: verifying $t_\varepsilon(w) \leq c_\varepsilon(w)$.** Next we check that the claim $t_\varepsilon(w) \leq c_\varepsilon(w)$ must hold. First, by definition of $c_\varepsilon(w)$, we have

$$\varepsilon^2 = \left( \sum_{j \in [L]} (w_j - c_\varepsilon(w))_+ \right)^2 + \sum_{j \in [L]} (w_j - c_\varepsilon(w))_+^2 \leq 2 \left( \sum_{j \in [L]} (w_j - c_\varepsilon(w))_+ \right)^2,$$

where the inequality holds by the properties of the $\ell_1$ and $\ell_2$ norms. Therefore,

$$\sum_{j \in [L]} (w_j - c)_+ \geq \varepsilon/\sqrt{2},$$

which verifies that

$$t_\varepsilon(w) = c_\varepsilon(w) + \varepsilon/\sqrt{2} - \sum_{j \in [L]} (w_j - c_\varepsilon(w))_+ \leq c_\varepsilon(w).$$

**Step 3: checking monotonicity.** Now we turn to verifying the monotonicity properties of $t_\varepsilon(w)$ and $c_\varepsilon(w)$. First we check that these functions are nonincreasing in $\varepsilon$. First, since $f_w(c)$ is nonincreasing in $c$, and $c_\varepsilon(w)$ is the solution to $f_w(c) = \varepsilon^2$, this immediately implies that $c_\varepsilon(w)$ is nonincreasing in $\varepsilon$. Next we turn to $t_\varepsilon(w)$. Define

$$t'(c) = c + \frac{1}{\sqrt{2}} \sqrt{ \left( \sum_{j \in [L]} (w_j - c)_+ \right)^2 + \sum_{j \in [L]} (w_j - c)_+^2 } - \sum_{j \in [L]} (w_j - c)_+,$$

so that we have $t_\varepsilon(w) = t'(c_\varepsilon(w))$ by construction. We can verify that $c \mapsto t'(c)$ is nondecreasing, and therefore, $t_\varepsilon(w) = t'(c_\varepsilon(w)) \leq t'(c_{\varepsilon'}(w)) = t_{\varepsilon'}(w)$, where the inequality holds since $c_\varepsilon(w) \leq c_{\varepsilon'}(w)$.

Next we check that $t_\varepsilon(w)$ and $c_\varepsilon(w)$ are nondecreasing functions of $w$. Fix any $w \leq w'$ (where the inequality is coordinatewise, i.e., $w_j \leq w'_j$ for all $j \in [L]$). Since $w \mapsto f_w(c)$ is a nondecreasing function, we have $f_w(c) \leq f_{w'}(c)$ for all $c$. We therefore have $f_{w'}(c_\varepsilon(w)) \geq f_w(c_\varepsilon(w)) = \varepsilon^2 = f_{w'}(c_\varepsilon(w'))$. Since $c \mapsto f_{w'}(c)$ is nonincreasing, therefore, $c_\varepsilon(w') \geq c_\varepsilon(w)$. Next we consider $t_\varepsilon$. Let $t''_\varepsilon(c) = c + \varepsilon/\sqrt{2} - \sum_{j \in [L]} (w_j - c)_+$, which is a nondecreasing function of $c$. Then we have $t_\varepsilon(w) = t''_\varepsilon(c_\varepsilon(w)) \leq t''_\varepsilon(c_\varepsilon(w')) = t_\varepsilon(w')$.

**Step 4: returning to the inflated argmax.** Finally, fixing any $w \in \mathbb{R}^L$, we will prove that $\text{argmax}^\varepsilon(w) = \{j : w_j > t_\varepsilon(w)\}$. First choose any $j \in [L]$ with $w_j > t_\varepsilon(w)$. We need to verify that $j \in \text{argmax}^\varepsilon(w)$. Define $v \in \mathbb{R}^L$ with entries

$$v_k = \begin{cases} \max\{c_\varepsilon(w) + \varepsilon/\sqrt{2}, w_j\}, & k = j, \\ \min\{c_\varepsilon(w), w_k\}, & k \neq j. \end{cases}$$

By construction, we have $v \in R_j^\varepsilon$. We calculate

$$
\begin{aligned}
\operatorname{dist}(w, R_j^\varepsilon)^2 &\leq \|w - v\|^2 \\
&= \left(w_j - \max\{c_\varepsilon(w) + \varepsilon/\sqrt{2}, w_j\}\right)^2 + \sum_{k \neq j} (w_k - \min\{c_\varepsilon(w), w_k\})^2 \\
&= \left(c_\varepsilon(w) + \varepsilon/\sqrt{2} - w_j\right)_+^2 + \sum_{k \neq j}(w_k - c_\varepsilon(w))_+^2 \\
&< \left(c_\varepsilon(w) + \varepsilon/\sqrt{2} - t_\varepsilon(w)\right)^2 + \sum_{k \neq j}(w_k - c_\varepsilon(w))_+^2 \\
&\leq \left(\sum_{k \in [L]}(w_k - c_\varepsilon(w))_+\right)^2 + \sum_{k \in [L]}(w_k - c_\varepsilon(w))_+^2 = \varepsilon^2,
\end{aligned}
$$

where the last two steps hold by definition of $t_\varepsilon(w)$ and $c_\varepsilon(w)$, while the strict inequality holds since $t_\varepsilon(w) \leq c_\varepsilon(w) < c_\varepsilon(w) + \varepsilon/\sqrt{2}$ as established above, while $w_j > t_\varepsilon(w)$ by assumption. Therefore $j \in \operatorname{argmax}^\varepsilon(w)$.

Now we check the converse. For this last step, we will need a lemma. The following result characterizes the projection of $w$ to the set $R_j^\varepsilon$.

**Lemma 16.** *Fix any $w \in \mathbb{R}^L$ and any $j \in [L]$. Then there is a unique $a \in \mathbb{R}$ satisfying*

$$
w_j = a + \varepsilon/\sqrt{2} - \sum_{k \neq j}(w_k - a)_+. \tag{11}
$$

*Moreover, defining the vector $v \in \mathbb{R}^L$ as*

$$
v_k = \begin{cases} a + \varepsilon/\sqrt{2}, & k = j, \\ a \wedge w_k, & k \neq j, \end{cases} \tag{12}
$$

*it holds that $v = \operatorname{argmin}_{u \in R_j^\varepsilon} \|w - u\|$, i.e., $v$ is the projection of $w$ to the set $R_j^\varepsilon$.*

Next suppose $w_j \leq t_\varepsilon(w)$. We need to verify that $j \notin \operatorname{argmax}^\varepsilon(w)$. Let $a$ and $v$ be defined as in Lemma 16 above. We can compare the equation (11) to the definition of $t_\varepsilon(w)$,

$$
t_\varepsilon(w) = c_\varepsilon(w) + \varepsilon/\sqrt{2} - \sum_{k \in [L]}(w_k - c_\varepsilon(w))_+ = c_\varepsilon(w) + \varepsilon/\sqrt{2} - \sum_{k \neq j}(w_k - c_\varepsilon(w))_+,
$$

where the last step holds since $w_j \leq t_\varepsilon(w)$ by assumption, and $t_\varepsilon(w) \leq c_\varepsilon(w)$ as proved above. Since $c \mapsto c + \varepsilon/\sqrt{2} - \sum_{k \neq j}(w_k - c)_+$ is an increasing function, and $w_j \leq t_\varepsilon(w)$, this implies

$a \le c_\varepsilon(w)$. We then calculate

$$\text{dist}(w, R_j^\varepsilon)^2 = \|w - v\|^2$$

$$= \left(w_j - \left(a + \varepsilon/\sqrt{2}\right)\right)^2 + \sum_{k \neq j}(w_k - a \wedge w_k)^2 \text{ by (12)}$$

$$= \left(w_j - \left(a + \varepsilon/\sqrt{2}\right)\right)^2 + \sum_{k \neq j}(w_k - a)_+^2$$

$$= \left(\sum_{k \neq j}(w_k - a)_+\right)^2 + \sum_{k \neq j}(w_k - a)_+^2 \text{ by (11)}$$

$$\geq \left(\sum_{k \neq j}(w_k - c_\varepsilon(w))_+\right)^2 + \sum_{k \neq j}(w_k - c_\varepsilon(w))_+^2 \text{ since } a \le c_\varepsilon(w)$$

$$= \left(\sum_{k \in [L]}(w_k - c_\varepsilon(w))_+\right)^2 + \sum_{k \in [L]}(w_k - c_\varepsilon(w))_+^2 \text{ since } w_j \le t_\varepsilon(w) \le c_\varepsilon(w)$$

$$= \varepsilon^2 \text{ by definition of } c_\varepsilon(w).$$

Therefore, $\text{dist}(w, R_j^\varepsilon) \ge \varepsilon$, and so $j \notin \text{argmax}^\varepsilon(w)$.

### A.6 Proofs of technical lemmas

#### A.6.1 Proof of Lemma 14

First, since $y$ is constrained to have nonnegative entries,

$$\|x + y + r\mathbf{1}_k\|^2 \ge \|(x + r\mathbf{1}_k)_+\|^2,$$

where for a vector $v = (v_1, \ldots, v_L) \in \mathbb{R}^L$, we write $(v)_+$ to denote the vector with $j$th entry given by $(v_j)_+ = \max\{v_j, 0\}$ for each $j$. The analogous bound holds for $\|x' + y' + r'\mathbf{1}_{k'}\|^2$.

$$\|x + y + r\mathbf{1}_k\|^2 + \|x' + y' + r'\mathbf{1}_{k'}\|^2 \ge \|(x + r\mathbf{1}_k)_+\|^2 + \|(x' + r'\mathbf{1}_{k'})_+\|^2.$$

We now need to show that

$$\|(x + r\mathbf{1}_k)_+\|^2 + \|(x' + r'\mathbf{1}_{k'})_+\|^2 \ge \varepsilon^2.$$

Next let

$$\bar{r} = \frac{r + r'}{2} = \frac{x^\top \mathbf{1}_k + x'^\top \mathbf{1}_{k'} - \varepsilon\sqrt{2}}{2}, \quad \Delta = \frac{-r + r'}{2},$$

so that we can write

$$r = \bar{r} - \Delta, \ r' = \bar{r} + \Delta$$

for some $\Delta \in \mathbb{R}$. We we therefore need to show $\inf_{\Delta \in \mathbb{R}} f(\Delta) \ge \varepsilon^2$, where

$$f(\Delta) = \|(x + (\bar{r} - \Delta)\mathbf{1}_k)_+\|^2 + \|(x' + (\bar{r} + \Delta)\mathbf{1}_{k'})_+\|^2.$$

First, we observe that we can restrict the range of $\Delta$. Specifically, for any $\Delta \ge \max_{j \in [L]} x_j + \bar{r}$, we have $f(\Delta) = \|(x' + (\bar{r} + \Delta)\mathbf{1}_{k'})_+\|^2$, which is a nondecreasing function; therefore,

$$\inf_{\Delta \ge \max_{j \in [L]} x_j + \bar{r}} f(\Delta) = f\left(\max_{j \in [L]} x_j + \bar{r}\right),$$

meaning that we do not need to consider values of $\Delta$ beyond this upper bound. Applying a similar argument for a lower bound, we see that from this point on we only need to verify that

$$f(\Delta) \ge \varepsilon^2 \text{ for} - \max_{j \in [k']} x'_j - \bar{r} \le \Delta \le \max_{j \in [L]} x_j + \bar{r}.$$

Moreover, for any $\Delta$, we have

$$(x_j + (\bar{r} - \Delta))^2 = (x_j + (\bar{r} - \Delta))_+^2 + (x_j + (\bar{r} - \Delta))_-^2 \le (x_j + (\bar{r} - \Delta))_+^2 + (\bar{r} - \Delta)^2$$

for all $j$, by nonnegativity of $x$. Furthermore we must have
$$(x_j + (\bar{r} - \Delta))^2 = (x_j + (\bar{r} - \Delta))_+^2$$
for at least one $j \in [L]$ when $\Delta \le \max_{j \in [L]} x_j + \bar{r}$ as specified above (i.e., because for $j$ maximizing the entry $x_j$, the value $(x + (\bar{r} - \Delta)\mathbf{1}_k)_j$ is nonnegative). Therefore,
$$\|(x + (\bar{r} - \Delta)\mathbf{1}_k)_+\|^2 \ge \|x + (\bar{r} - \Delta)\mathbf{1}_k\|^2 - (k-1)(\bar{r} - \Delta)^2.$$
Similarly, we can show that
$$\|(x' + (\bar{r} + \Delta)\mathbf{1}_{k'})_+\|^2 \ge \|x' + (\bar{r} + \Delta)\mathbf{1}_{k'}\|^2 - (k'-1)(\bar{r} + \Delta)^2.$$
We then calculate
$$\begin{aligned}
f(\Delta) &= \|(x + (\bar{r} - \Delta)\mathbf{1}_k)_+\|^2 + \|(x' + (\bar{r} + \Delta)\mathbf{1}_{k'})_+\|^2 \\
&\ge \|x + (\bar{r} - \Delta)\mathbf{1}_k\|^2 - (k-1)(\bar{r} - \Delta)^2 + \|x' + (\bar{r} + \Delta)\mathbf{1}_{k'}\|^2 - (k'-1)(\bar{r} + \Delta)^2 \\
&= \|x\|^2 + 2(\bar{r} - \Delta)x^\top \mathbf{1}_k + (\bar{r} - \Delta)^2 + \|x'\|^2 + 2(\bar{r} + \Delta)x'^\top \mathbf{1}_{k'} + (\bar{r} + \Delta)^2 \\
&= 2\varepsilon^2 - (x^\top \mathbf{1}_k)^2 - (x'^\top \mathbf{1}_{k'})^2 + 2(\bar{r} - \Delta)x^\top \mathbf{1}_k + 2(\bar{r} + \Delta)x'^\top \mathbf{1}_{k'} + 2\bar{r}^2 + 2\Delta^2,
\end{aligned}$$
where the last step holds by (8).

Writing $z = x^\top \mathbf{1}_k$ and $z' = x'^\top \mathbf{1}_{k'}$ for convenience, we have
$$\begin{aligned}
f(\Delta) &\ge 2\varepsilon^2 - z^2 - z'^2 + 2(\bar{r} - \Delta)z + 2(\bar{r} + \Delta)z' + 2\bar{r}^2 + 2\Delta^2 \\
&= 2\varepsilon^2 - z^2 - z'^2 + 2\left(\frac{z + z' - \varepsilon\sqrt{2}}{2} - \Delta\right)z + 2\left(\frac{z + z' - \varepsilon\sqrt{2}}{2} + \Delta\right)z' \\
&\qquad + 2\left(\frac{z + z' - \varepsilon\sqrt{2}}{2}\right)^2 + 2\Delta^2 \\
&= \varepsilon^2 + 4\left(z - \varepsilon/\sqrt{2}\right)\left(z' - \varepsilon/\sqrt{2}\right) + 2\left(\Delta - \frac{z - z'}{2}\right)^2 \\
&\ge \varepsilon^2 + 4\left(z - \varepsilon/\sqrt{2}\right)\left(z' - \varepsilon/\sqrt{2}\right).
\end{aligned}$$
where the second step holds because $\bar{r} = \frac{r + r'}{2} = \frac{z + z' - \varepsilon\sqrt{2}}{2}$ by (9).

Finally, we also have $x^\top \mathbf{1}_k = \|x\|_1 \ge \|x\|$, where the first step holds since $x$ is nonnegative, and so by assumption (8) we have $z = x^\top \mathbf{1}_k \ge \varepsilon/\sqrt{2}$. Similarly, $z' \ge \varepsilon/\sqrt{2}$. Therefore, $4\left(z - \varepsilon/\sqrt{2}\right)\left(z' - \varepsilon/\sqrt{2}\right) \ge 0$, and so we have shown that $f(\Delta) \ge \varepsilon^2$, as desired.

### A.6.2   Proof of Lemma 15

First, suppose $w \in R_j^\varepsilon$. Then $\operatorname{argmax}(w) = \{j\}$, and so $j \in \operatorname{argmax}^\varepsilon(w)$ by Proposition 10. Next, let $c = w_j - \varepsilon/\sqrt{2}$. Then $w_\ell \le c$ for all $\ell \ne j$, and so defining $f_w$ as in (10), we have
$$f_w(c) = (w_j - c)^2 + (w_j - c)^2 = \varepsilon^2 \implies c_\varepsilon(w) = c = w_j - \varepsilon/\sqrt{2}.$$
Then
$$t_\varepsilon(w) = c_\varepsilon(w) + \varepsilon/\sqrt{2} - \sum_{\ell \in [L]} (w_\ell - c_\varepsilon(w))_+ = \left(w_j - \varepsilon/\sqrt{2}\right) + \varepsilon/\sqrt{2} - \varepsilon/\sqrt{2} = w_j - \varepsilon/\sqrt{2},$$
and so for all $\ell \ne j$, we have $w_\ell \le w_j - \varepsilon/\sqrt{2} = t_\varepsilon(w)$ and thus $\ell \notin \operatorname{argmax}^\varepsilon(w)$. This verifies that $\operatorname{argmax}^\varepsilon(w) = \{j\}$.

To prove the converse, suppose $\operatorname{argmax}^\varepsilon(w) = \{j\}$. By Proposition 10, we then have $\operatorname{argmax}(w) = \{j\}$. Let $k \in \operatorname{argmax}_{\ell \ne j} w_\ell$, then to show $w \in R_j^\varepsilon$ it suffices to show that $w_j \ge w_k + \varepsilon/\sqrt{2}$. Define $v \in \mathbb{R}^L$ as
$$v_\ell = \begin{cases} w_k, & \ell = j, \\ w_k + \varepsilon/\sqrt{2}, & \ell = k, \\ w_\ell, & \ell \ne j, k. \end{cases}$$
Then $v \in R_k^\varepsilon$, and so we must have $\|w - v\| \ge \varepsilon$ since $k \notin \operatorname{argmax}^\varepsilon(w)$. We calculate
$$\|w - v\|^2 = (w_j - v_j)^2 + (w_k - v_k)^2 = (w_j - w_k)^2 + (\varepsilon/\sqrt{2})^2.$$
This implies that we must have $(w_j - w_k)^2 \ge \varepsilon^2/2$, which completes the proof.

### A.6.3 Proof of Lemma 16

First we verify existence and uniqueness of $a$. Let

$$f(a) = w_j - a + \sum_{k \neq j} (w_k - a)_+.$$

Then $f : \mathbb{R} \to \mathbb{R}$ is a continuous and strictly decreasing bijection, so $f(a) = \varepsilon/\sqrt{2}$ must have a unique solution.

Next let $a$ be this unique solution and let $v$ be defined as in (12). Now we verify that $v$ is the unique solution to the optimization problem

$$v = \operatorname{argmin}_{u \in \mathbb{R}^L} \left\{ \frac{1}{2} \|u - w\|^2 : u_j \geq u_k + \varepsilon/\sqrt{2} \text{ for all } k \neq j \right\},$$

which defines the projection of $w$ to $R_j^\varepsilon$. By the Karush–Kuhn–Tucker (KKT) conditions for first-order optimality, it is sufficient to verify that

$$v - w - \sum_{k \neq j} \lambda_k (\mathbf{e}_j - \mathbf{e}_k) = 0, \tag{13}$$

where $\lambda_k \geq 0$ for all $k$, and $\lambda_k = 0$ for any inactive constraints, i.e., any $k$ for which $v_j > v_k + \varepsilon/\sqrt{2}$. Let $\lambda_k = (w_k - a)_+$ for each $k \neq j$. Note that if $v_j > v_k + \varepsilon/\sqrt{2}$ (i.e., an inactive constraint), then we must have $w_k < a$ and so $\lambda_k = 0$, by construction of $v$. Then, for any $\ell \neq j$,

$$\left( v - w - \sum_{k \neq j} \lambda_k (\mathbf{e}_j - \mathbf{e}_k) \right)_\ell = v_\ell - w_\ell + \lambda_\ell = a \wedge w_\ell - w_\ell + (w_\ell - a)_+ = 0,$$

where $\mathbf{e}_k$ denotes the $k$th canonical basis vector in $\mathbb{R}^L$. This proves that the $\ell$th coordinate of the system of equations (13) holds for each $\ell \neq j$. Now we verify the $j$th coordinate. We have

$$\left( v - w - \sum_{k \neq j} \lambda_k (\mathbf{e}_j - \mathbf{e}_k) \right)_j = v_j - w_j - \sum_{k \neq j} \lambda_k$$

$$= a + \varepsilon/\sqrt{2} - w_j - \sum_{k \neq j} (w_k - a)_+ = \varepsilon/\sqrt{2} - f(a) = 0,$$

by our choice of $a$ as the solution to $f(a) = \varepsilon/\sqrt{2}$. This verifies the KKT conditions, and thus, $v$ is the projection of $w$ to $R_j^\varepsilon$ as claimed.

## B  Extension to randomized algorithms

As mentioned in Section 2, in many applications it is common to use randomization in the construction of a classification procedure, in the learning algorithm $\mathcal{A}$ and/or in the selection rule $s$. In this section we formalize this more general framework, and will see how our results apply.

First, in the non-random setting, a learning algorithm $\mathcal{A}$ maps a data set $\mathcal{D} \in (\mathcal{X} \times [L])^n$ to a fitted probability estimate function $\hat{p} : \mathcal{X} \to \Delta_{L-1}$—we write the regression procedure as $\hat{p} = \mathcal{A}(\mathcal{D})$. In the randomized setting, we also allow for external randomness, $\hat{p} = \mathcal{A}(\mathcal{D}; \xi)$, where $\xi \sim \text{Uniform}[0, 1]$ is a random seed (e.g., we might use $\xi$ to randomly shuffle the training data when running stochastic gradient descent).

Next, in the non-random setting, a selection rule $s$ is a map from $\Delta_{L-1}$ to subsets of $[L]$, resulting in a candidate set of labels $\hat{S} = s(\hat{p}(x))$. In the randomized setting (for instance, if $s$ is the argmax but with ties broken at random), we instead include a random seed $\zeta \sim \text{Uniform}[0, 1]$, and write $\hat{S} = s(\hat{p}(x); \zeta)$. Formally, then, we have $s : \Delta_{L-1} \times [0, 1] \to \wp([L])$. (Note that the selection rule proposed in this work—the inflated argmax—is itself *not* random.)

Combining the two stages of the procedure, then, the classification algorithm is given by $\mathcal{C} = s \circ \mathcal{A}$, where given a training set $\mathcal{D}$ and a test point $x$, along with i.i.d. random seeds $\xi, \zeta \sim \text{Uniform}[0,1]$, we return the candidate set of labels given by

$$\mathcal{C}(\mathcal{D}, x; \xi, \zeta) = s(\hat{p}(x); \zeta) \text{ where } \hat{p} = \mathcal{A}(\mathcal{D}; \xi).$$

Of course, we can derive the non-random setting as a special case, simply by designing $\mathcal{A}$ and $s$ to not depend on the random seeds $\xi$ and $\zeta$.

Now, how do the results of this paper extend to this randomized setting? First, we need to reconsider our definition of selection stability (Definition 2). We will now define it as

$$\frac{1}{n} \sum_{i=1}^{n} \Pr \left\{ \hat{S} \cap \hat{S}^{\setminus i} = \varnothing \right\} \le \delta,$$

where $\hat{S} = \mathcal{C}(\mathcal{D}, x; \xi, \zeta)$ and $\hat{S}^{\setminus i} = \mathcal{C}(\mathcal{D}^{\setminus i}, x; \xi, \zeta)$, and where the probabilities are taken with respect to the distribution of the random seeds. We can also consider a notion of algorithmic stability (Definition 3) that allows for randomization: we say that $\mathcal{A}$ has tail stability $(\varepsilon, \delta)$ if

$$\frac{1}{n} \sum_{i=1}^{n} \Pr \left\{ \|\hat{p}(x) - \hat{p}^{\setminus i}(x)\| \ge \varepsilon \right\} \le \delta,$$

where $\hat{p} = \mathcal{A}(\mathcal{D}; \xi)$ and $\hat{p}^{\setminus i} = \mathcal{A}(\mathcal{D}^{\setminus i}; \xi)$, and where again probability is taken with respect to the distribution of $\xi$.

With these definitions in place, we observe that the result of Proposition 5 still holds in this more general setting for the selection rule $\text{argmax}^{\varepsilon}$: if $\mathcal{A}$ is a randomized algorithm with tail stability $(\varepsilon, \delta)$ (under the new definition given above), then the randomized classification algorithm given by $\mathcal{C} = \text{argmax}^{\varepsilon} \circ \mathcal{A}$ has selection stability $\delta$ (again, under the new definition given above), simply due to the $\varepsilon$-compatibility property of $\text{argmax}^{\varepsilon}$ (Theorem 9). [SBW24b] proved Theorem 8 in this more general setting of randomized learning algorithms $\mathcal{A}$, so Theorem 6 holds for randomized $\mathcal{A}$ under this more general definition of selection stability.

## C Extension to finitely many bags

In Section 3.1, we discussed how in practice, the bagged algorithm $\widetilde{\mathcal{A}}_m$ would be constructed by taking an empirical average over some large number $B$ of sampled bags, rather than computing the exact expectation $\mathbb{E}_r[\mathcal{A}(\mathcal{D}^r)(x)]$. Now that we have allowed for randomized learning algorithms (as in Appendix B), we can formalize this setting: for a random seed $\xi \sim \text{Uniform}[0,1]$, we define

$$\widetilde{\mathcal{A}}_m(\mathcal{D}, \xi)(x) = \frac{1}{B} \sum_{b=1}^{B} \mathcal{A}(\mathcal{D}^{r_b})(x),$$

where the random draw of $B$ many bags, $r_1, \ldots, r_B$, is generated using the random seed $\xi$. Recall that [SBW24b] showed tail stability for the bagged version of any algorithm (as restated in Theorem 8)—their work also gives a result for the finite-$B$ case, as follows:

**Theorem 17** ([SBW24b])**.** *For any base learning algorithm $\mathcal{A}$ returning outputs in $\Delta_{L-1}$, the bagged algorithm $\widetilde{\mathcal{A}}_m$ (computed with a finite number of bags, $B$) has tail stability $(\varepsilon, \delta)$ for any $\varepsilon, \delta > 0$ satisfying*

$$\delta = \varepsilon^{-2} \cdot (1 - 1/L) \left( \frac{1}{n-1} \cdot \frac{p_{n,m}}{1 - p_{n,m}} + \frac{16e^2}{B} \right).$$

In particular, combined with our result on the $\varepsilon$-compatibility of the inflated argmax, we have the following generalization of our main result (Theorem 6):

**Theorem 18.** *Fix any sample size $n$, any bag size $m$, any inflation parameter $\varepsilon > 0$, and any number of bags $B \ge 1$. For any base learning algorithm $\mathcal{A}$, the classification algorithm $\mathcal{C} = \text{argmax}^{\varepsilon} \circ \widetilde{\mathcal{A}}_m$, obtained by combining the bagged version of $\mathcal{A}$ (with $B$ many bags) together with the inflated argmax, satisfies selection stability $\delta$ where*

$$\delta = \varepsilon^{-2} \cdot (1 - 1/L) \left( \frac{1}{n-1} \cdot \frac{p_{n,m}}{1 - p_{n,m}} + \frac{16e^2}{B} \right). \tag{14}$$

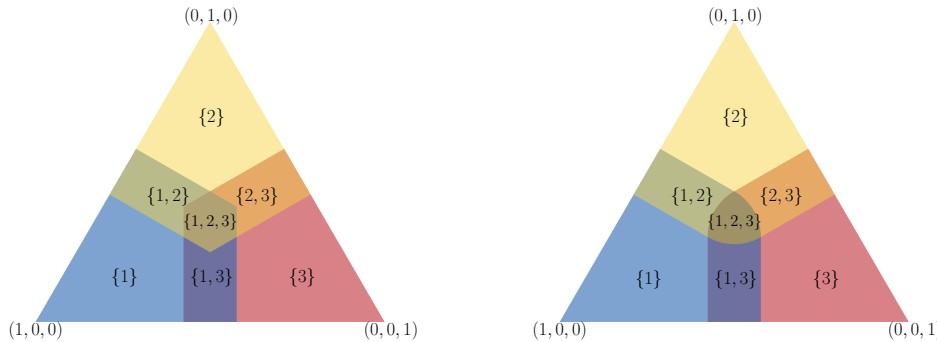

Figure 3: The fixed-margin selection rule (3), left, and the inflated argmax (1), right, when $L = 3$.

Comparing to our main result for the derandomized case (Theorem 6, which can be interpreted as taking $B \to \infty$), we see that this finite-$B$ result is essentially the same as Theorem 6 once $B \gg n$.

## D Compare to a simpler selection rule: the fixed-margin rule

In this section, we compare to the simpler fixed-margin selection rule, which was defined earlier in (3)—for convenience, we define it again here:

$$s^\varepsilon_{\text{margin}}(w) = \left\{ j \in [L] : w_j > \max_{\ell \in [L]} w_\ell - \varepsilon/\sqrt{2} \right\},$$

the set of all indices $j$ that are within some margin of being the maximum.

In this section, we will see that this fixed-margin selection rule is $\varepsilon$-compatible. Since this rule clearly has the advantage of being much simpler than the inflated argmax (both in terms of its definition and interpretability, and in terms of its computation), we might ask whether this rule is perhaps preferable to the more complex inflated argmax. However, we will also see that the fixed-margin selection rule can be very inefficient compared to the inflated argmax: the inflated argmax can never return a larger set, and will often return a substantially smaller one.

### D.1 Theoretical results

First, we verify that this simple rule satisfies $\varepsilon$-compatibility (Definition 4).

**Proposition 19.** *For any $\varepsilon > 0$, the selection rule $s^\varepsilon_{\text{margin}}$ is $\varepsilon$-compatible.*

In particular, since $s^\varepsilon_{\text{margin}}$ clearly contains the argmax (i.e., $\operatorname{argmax}(w) \subseteq s^\varepsilon_{\text{margin}}(w)$) and satisfies permutation invariance (in the sense of Proposition 10), by Proposition 12 this immediately implies that

$$s^\varepsilon_{\text{margin}}(w) = \{j\} \implies \operatorname{argmax}^\varepsilon(w) = \{j\}$$

for all $w \in \mathbb{R}^L$, i.e., $\operatorname{argmax}^\varepsilon$ is at least as good as $s^\varepsilon_{\text{margin}}$ at returning a singleton set. However, for this particular selection rule, we can state an even stronger result:

**Proposition 20.** *For any $w \in \mathbb{R}^L$ and any $\varepsilon > 0$,*

$$s^\varepsilon_{\text{margin}}(w) \supseteq \operatorname{argmax}^\varepsilon(w).$$

In particular, this theoretical result ensures that the set of candidate labels $\hat{S}$ returned by the inflated argmax can never be larger than for the fixed-margin selection rule. In low dimensions, however, the improvement is small. Indeed, for $L = 2$, we actually have $s^\varepsilon_{\text{margin}}(w) = \operatorname{argmax}^\varepsilon(w)$ for all $w$. For $L = 3$, Figure 3 shows that the inflated argmax is strictly better (i.e., the set inclusion result of Proposition 20 can, for certain values of $w$, be a strict inclusion), but the difference in this low-dimensional setting appears minor. Next, however, we will see empirically that as the dimension $L$ grows, the benefit of the inflated argmax can be substantial.

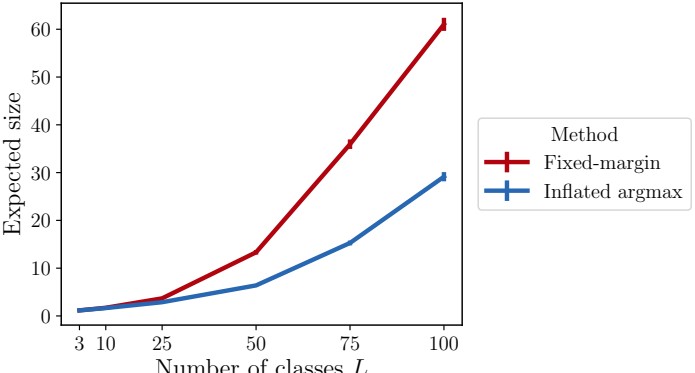

Figure 4: Simulation to compare the selection rules $\mathrm{argmax}^\varepsilon$ and $s^\varepsilon_{\mathrm{margin}}$. The figure shows the average set size, $|\mathrm{argmax}^\varepsilon(w)|$ and $|s^\varepsilon_{\mathrm{margin}}(w)|$, averaged over $1{,}000$ random draws of $w$ (with standard error bars shown). See Appendix D.2 for details.

## D.2  Simulation to compare the selection rules

In this section, we compare the fixed-margin rule to the inflated argmax using randomly generated probability weights. For our simulation, we consider the size of the two sets for various values of $L$. To sample the weights, we draw a standard Gaussian random vector $Z \sim \mathcal{N}(0, \mathbf{I}_L)$ and define $w$ as the softmax of $Z$, i.e.,

$$w_j = \frac{e^{Z_j}}{\sum_{\ell \in [L]} e^{Z_\ell}}$$

for $j \in [L]$. We then compute $\mathrm{argmax}^\varepsilon(w)$ and $s^\varepsilon_{\mathrm{margin}}(w)$, where $\varepsilon = 0.1$. In Figure 4, we present results comparing the average size of the sets returned by each of the two methods. In this setting, we see that the inflated argmax has substantial gains over the fixed-margin selection rule when the number of classes $L$ is large. Even when $L = 25$, the ratio of the expected sizes is about $78\%$, so the inflated argmax has a nontrivial advantage in that setting. When $L = 100$, the ratio of expected sizes is $48\%$, meaning the inflated argmax is on average more than twice as small.

## D.3  Proofs for the fixed-margin selection rule

### D.3.1  Proof of Proposition 19

Let $w, v \in \mathbb{R}^L$ with $s^\varepsilon_{\mathrm{margin}}(w) \cap s^\varepsilon_{\mathrm{margin}}(v) = \varnothing$. Let $j \in \mathrm{argmax}_\ell w_\ell$ and $k \in \mathrm{argmax}_\ell v_\ell$. Then clearly $j \in s^\varepsilon_{\mathrm{margin}}(w)$, so since $s^\varepsilon_{\mathrm{margin}}(w) \cap s^\varepsilon_{\mathrm{margin}}(v) = \varnothing$, this implies $j \notin s^\varepsilon_{\mathrm{margin}}(v)$. By definition, then, $v_j \leq \max_\ell v_\ell - \varepsilon/\sqrt{2} = v_k - \varepsilon/\sqrt{2}$. By an identical argument, we have $w_k \leq w_j - \varepsilon/\sqrt{2}$. Then

$$\begin{aligned}
\|w - v\|^2 &\geq (w_j - v_j)^2 + (w_k - v_k)^2 \\
&\geq (w_j - v_j)^2_+ + (v_k - w_k)^2_+ \\
&\geq \left(w_j - (v_k - \varepsilon/\sqrt{2})\right)^2_+ + \left(v_k - (w_j - \varepsilon/\sqrt{2})\right)^2_+ \\
&\geq \inf_{t \in \mathbb{R}} \left\{ (\varepsilon/\sqrt{2} - t)^2_+ + (\varepsilon/\sqrt{2} + t)^2_+ \right\} \\
&= 2(\varepsilon/\sqrt{2})^2 = \varepsilon^2.
\end{aligned}$$

This proves $\|w - v\| \geq \varepsilon$, and therefore, we have shown that $\varepsilon$-compatibility is satisfied.

### D.3.2 Proof of Proposition 20

Suppose $j \in \operatorname{argmax}^\varepsilon(w)$. Then $\operatorname{dist}(w, R_j^\varepsilon) < \varepsilon$, so we can find some $v \in R_j^\varepsilon$ with $\|w - v\| < \varepsilon$. Then, for any $k \neq j$,

$$
\begin{aligned}
w_j - w_k &= v_j - v_k + (w_j - v_j) + (v_k - w_k) \\
&\geq \varepsilon/\sqrt{2} + (w_j - v_j) + (v_k - w_k) \text{ since } v \in R_j^\varepsilon \\
&\geq \varepsilon/\sqrt{2} - \left( |w_j - v_j| + |w_k - v_k| \right) \\
&\geq \varepsilon/\sqrt{2} - \sqrt{2}\sqrt{(w_j - v_j)^2 + (w_k - v_k)^2} \\
&\geq \varepsilon/\sqrt{2} - \sqrt{2}\|w - v\| \\
&> \varepsilon/\sqrt{2} - \sqrt{2} \cdot \varepsilon \\
&= -\varepsilon/\sqrt{2}.
\end{aligned}
$$

Since this holds for all $k \neq j$, then,

$$
w_j > \max_{k \in [L]} w_k - \varepsilon/\sqrt{2},
$$

which proves that $j \in s_{\mathrm{margin}}^\varepsilon(w)$.

## E   Additional experimental results

In this section, we extend our experiment from Section 4 to consider some additional existing methods as baselines and evaluate each method using some common metrics from set-valued classification.

**Selection rules.**   We compare the following selection rules:

1. Standard argmax.

2. $\varepsilon$-inflated argmax with tolerance $\varepsilon = .05$.

3. Top-$K$ classification with $K = 2$.

4. Thresholding: including classes in the output set until the sum of probabilities becomes at least $\tau = 0.8$, i.e.,

$$
\Gamma_\tau^*(w) := \left\{ \ell \in [L] : w_\ell \geq w_{[\hat{k}]} \right\}, \text{ where}
$$
$$
\hat{k} = \min\{k : w_{[1]} + \cdots + w_{[k]} \geq \tau\}.
$$

5. Nondeterministic classification optimized for $F_1$-score [DDB09]:

$$
\mathrm{NDC}_{F_1}(w) := \left\{ \ell \in [L] : w_\ell \geq w_{[\hat{k}]} \right\}, \text{ where}
$$
$$
\hat{k} = \min\{k : w_{[1]} + \cdots + w_{[k]} \geq (k+1)w_{[k+1]}\},
$$

with the convention $w_{[L+1]} = 0$.

6. Set-valued Bayes-optimal prediction [MWDH21]: for any utility $u : [L] \times 2^{[L]} \setminus \{\varnothing\} \to \mathbb{R}_+$,

$$
\mathrm{SVBOP}_u(w) := \operatorname{argmax}_{S \subseteq [L]} \sum_{\ell \in [L]} w_\ell \cdot u(\ell, S).
$$

We consider two utility functions based on $u_{65}$ and $u_{80}$, defined below.

**Evaluation metrics.**   We evaluate each method based on a variety of metrics. In addition to $\beta_{\mathrm{prec}}$ and $\beta_{\mathrm{set\text{-}size}}$, defined in Section 4, we assess each method in terms of *utility-discounted predictive accuracy* [ZCM12]. For a set $S \subseteq [L]$ and label $\ell \in [L]$, define for some parameters $\alpha, \beta > 0$,

$$
u(\ell, S) := \mathbf{1}\{\ell \in S\} \cdot \left( \frac{\alpha}{|S|} - \frac{\beta}{|S|^2} \right).
$$

| Sel. rule | Algo. | $\beta_{\text{correct-single}}$ ↗ | $\beta_{\text{set-size}}$ ↘ | $u_{65}$ ↗ | $u_{80}$ ↗ | $\beta_{\text{sup. infl.}}$ ↘ |
|---|---|---|---|---|---|---|
| argmax | $\mathcal{A}$ | 0.879 (0.003) | 1.000 (0.000) | 0.879 (0.003) | 0.879 (0.003) | - |
| argmax | $\widetilde{\mathcal{A}}_m$ | 0.892 (0.003) | 1.000 (0.000) | 0.892 (0.003) | 0.893 (0.003) | - |
| argmax$^\varepsilon$ | $\mathcal{A}$ | 0.873 (0.003) | 1.015 (0.001) | 0.881 (0.003) | 0.883 (0.003) | 0.496 (0.005) |
| argmax$^\varepsilon$ | $\widetilde{\mathcal{A}}_m$ | 0.886 (0.003) | 1.017 (0.001) | 0.895 (0.003) | 0.897 (0.003) | 0.474 (0.005) |
| top-2 | $\mathcal{A}$ | 0.000 (0.000) | 2.000 (0.000) | 0.632 (0.001) | 0.777 (0.001) | 0.905 (0.003) |
| top-2 | $\widetilde{\mathcal{A}}_m$ | 0.000 (0.000) | 2.000 (0.000) | 0.632 (0.001) | 0.777 (0.001) | 0.918 (0.003) |
| $\Gamma^*_{0.8}$ | $\mathcal{A}$ | 0.756 (0.004) | 1.248 (0.005) | 0.880 (0.002) | 0.909 (0.002) | 0.622 (0.005) |
| $\Gamma^*_{0.8}$ | $\widetilde{\mathcal{A}}_m$ | 0.704 (0.005) | 1.356 (0.006) | 0.867 (0.002) | 0.907 (0.002) | 0.701 (0.005) |
| NDC for $F_1$ | $\mathcal{A}$ | 0.832 (0.004) | 1.105 (0.003) | 0.889 (0.003) | 0.902 (0.003) | 0.531 (0.005) |
| NDC for $F_1$ | $\widetilde{\mathcal{A}}_m$ | 0.825 (0.004) | 1.138 (0.004) | 0.898 (0.003) | 0.915 (0.002) | 0.586 (0.005) |
| SVBOP$_{u_{65}}$ | $\mathcal{A}$ | 0.838 (0.004) | 1.091 (0.003) | 0.889 (0.003) | 0.901 (0.003) | 0.525 (0.005) |
| SVBOP$_{u_{65}}$ | $\widetilde{\mathcal{A}}_m$ | 0.833 (0.004) | 1.119 (0.003) | 0.899 (0.003) | 0.915 (0.002) | 0.577 (0.005) |
| SVBOP$_{u_{80}}$ | $\mathcal{A}$ | 0.777 (0.004) | 1.190 (0.004) | 0.885 (0.003) | 0.910 (0.002) | 0.611 (0.005) |
| SVBOP$_{u_{80}}$ | $\widetilde{\mathcal{A}}_m$ | 0.744 (0.004) | 1.245 (0.005) | 0.882 (0.002) | 0.914 (0.002) | 0.690 (0.005) |

Table 2: Results on the Fashion MNIST data set. The table displays the frequency of returning the correct label as a singleton $\beta_{\text{correct-single}}$, average size $\beta_{\text{set-size}}$, utility-discounted predictive accuracies $u_{65}$ and $u_{80}$, and the superfluous inflation, $\beta_{\text{sup. infl.}}$. For each metric, the symbol ↗ indicates that higher values are desirable, while ↘ indicates that lower values are desirable. Results for the base algorithm are in white, and results for the subbagged algorithm are in gray. Standard errors are in parentheses.

We use the measures $u_{65}$ with $(\alpha, \beta) = (1.6, 0.6)$ and $u_{80}$ with $(\alpha, \beta) = (2.2, 1.2)$, which respectively give small and large rewards for being cautious [NDMH18].

Finally, we directly assess the extent to which each selection rule resorts to returning a non-singleton set on difficult instances. Specifically, we define the *superfluous inflation*, which, for a selection rule $s$, is the ratio of the accuracy of the standard argmax given $s$ returns at least two labels divided by the accuracy of $s$ given $s$ returns at least two labels:

$$\beta_{\text{sup. infl.}} := \frac{\sum_{j=1}^{N} \mathbf{1}\left\{\tilde{Y}_j \in \text{argmax}(\hat{p}(\tilde{X}_j)), |s(\hat{p}(\tilde{X}_j))| \geq 2\right\}}{\sum_{j=1}^{N} \mathbf{1}\left\{\tilde{Y}_j \in s(\hat{p}(\tilde{X}_j)), |s(\hat{p}(\tilde{X}_j))| \geq 2\right\}}.$$

If this ratio is close to 1, it means the standard argmax is correct as often as $s$ (when the latter expresses uncertainty by returning multiple labels), so outputting multiple labels could be seen as overly conservative. Note that the argmax never returns more than a singleton, so its superfluous inflation is left blank in our results.

**Results.** This experiment shows that the inflated argmax combined with the subbagged algorithm $\widetilde{\mathcal{A}}_m$ has accuracy (according to a variety of metrics) comparable with several alternative methods, does not output overly large sets of candidate labels, and at the same time admits rigorous stability guarantees. That is, our algorithmic framework does not unduly harm empirical performance.

In Table 2, we present results for each selection rule applied to the base algorithm $\mathcal{A}$ and the subbagged algorithm $\widetilde{\mathcal{A}}_m$ described in Section 4. The inflated argmax has significantly higher average precision and significantly smaller set sizes than all of the alternative set-valued classifiers. These two measures are related, since a selection rule can only have high precision if it frequently returns the correct label *as a singleton set*. The inflated argmax also has the lowest superfluous inflation, meaning that it tends return multiple labels on difficult instances. The $u_{65}$ of the inflated argmax is also within two standard errors of the $u_{65}$ of SVBOP$_{65}$, which seeks to optimize this utility. Our method does have a significantly lower $u_{80}$ than many of the competing methods, since this utility is more forgiving of returning multiple labels.

While this experiment considers many different perspectives on set-valued classification, we reiterate that our chief contribution is distribution-free stability guarantees. This means that, regardless of the dataset or base algorithm used, we can guarantee that our method will be stable. In the context of our experiment, Theorem 6 guarantees that $\delta_j \leq \delta^* = \varepsilon^{-2} \cdot \frac{1-1/L}{n-1} \cdot \frac{p_{n,m}}{1-p_{n,m}} \approx 0.006$ for *every* test point $j = 1, \ldots, N$. Furthermore, our optimality result shows that the inflated argmax returns a singleton as often as possible among all $\varepsilon$-compatible selection rules.

