# OpenReview forum: "Building a stable classifier with the inflated argmax"
_NeurIPS.cc/2024/Conference — NeurIPS 2024 poster_

### Official Review · Reviewer_LzW3 · 2024-07-08

**Soundness:** 3
**Presentation:** 3
**Contribution:** 3
**Rating:** 7
**Confidence:** 3

**Summary:**

This paper studies and proposes a new theoretical framework to study algorithmic stability for multiclass classification with a focus on a stable class selection given predicted class probability. Based on the proposed theory, the proposed selection criterion based on bagging called "inflated argmax" was proposed. Theoretical analysis suggests that the proposed inflated argmax is stable, while also give a relatively tight decision for the candidate prediction set. Simple experiments using Fashion-MNIST were also provided to show the stability of inflated argmax as well as how tight the candidate set can be.

**Strengths:**

1. The paper provides strong theoretical framework for future study of stable selection criteria for the multiclass classification problem given predicted class probability.
2. The proposed notion of $\epsilon$- compatible is simple to understand and a reasonable extension of classical algorithmic stability noted in Definition 2. I feel this notion is neat and also very useful for further analysis in Proposition 4 and beyond.
3. The simple rule that satisfies the stability but could give a looser quality than inflated argmax was also provided to give a better understanding of the problem. The study was quite detailed in Appendix D., which I find quite amazing.
4. Despite the complicated notion of inflated argmax, the efficient computation of selection rule once bagging is done is nicely provided in Proposition 10.

**Weaknesses:**

1. Experiments were quite weak in my opinion to highlight the effectiveness of inflated argmax. For example, more reasonable baseline that has larger than 1 $\beta_{\mathrm{size}}$ could have been provided to show that inflated argmax is quite favorable if it allows to predict more than one class. Curreent baselines are restrictive in the sense that only one class can be chosen (argmax). Several improvements could also be done (please see the questions section for possible improvements).
2. The proposed inflated argmax requires bagging, which can be quite computationally expensive, or prohibitive in some scenarios under limited resources or large dataset/model.

**Questions:**

1. Can we also see the performance of inflated argmax *without bagging* in the experiment? In my understanding, this should also be possible with the proposed efficient algorithm. This can be useful to see how important the bagging method is needed for the proposed algorithm.
2. Visualizing the number of bags needed to perform reasonably in the experiment can also be useful. Because it can be expensive if we use more sophisticated algorithm or using larger dataset.
3. In deep learning, overconfident problem of probability can often be an issue when one wants to use probability score. How does this problem affect the proposed inflated argmax in practice?
4. Can we know that it is likely that the second class to be chosen as a candidate class is likely to be the second-largest probability class?
5. Is it reasonable to say that precision is similar given different average precision and standard error provided (Lines 293--294)?
6. I was wondering if there are other possible baselines in related work that could be used in the experiments?
7. Suggestions on simple baselines
7.1 Top-2 class prediction, size of beta will be 2.
7.2 Predicting classes until sum of probability becomes a certain constant, e.g., if we set threshold = 0.8 and prediction probability vector is [0.1, 0.1, 0.1, 0.2, 0.6], then we will pick the last two classes (0.2 and 0.6). This simple idea looks intuitive and gives flexible set and this is the most simple baseline I had in mind that could be effective when I first read this paper. How do you think about this baseline?

I think the core contribution of this paper is the theory of stable classification rule. Therefore, although the experiment looks quite weak, my current rating is on the accept side (weak accept). I know it can be too much to ask to improve experiments a lot and I do not expect authors to fix everything according to my suggestions. But I would appreciate if the authors can reply to make sure that my understanding is correct.

**Limitations:**

Discussion of limitations of the proposed method and potential future work were appropriate.

---

> ### Author Rebuttal · Authors · 2024-08-06
>
> 1. *“Experiments were quite weak in my opinion to highlight the effectiveness of inflated argmax. For example, more reasonable baseline that has larger than 1 𝛽_size could have been provided to show that inflated argmax is quite favorable if it allows to predict more than one class. Curreent baselines are restrictive in the sense that only one class can be chosen (argmax). Several improvements could also be done (please see the questions section for possible improvements).”*
>
>     Please see discussion point 1 in our global response.
>
>
> 2. *“The proposed inflated argmax requires bagging, which can be quite computationally expensive, or prohibitive in some scenarios under limited resources or large dataset/model.”*
>
>     The inflated argmax operates on the predicted probabilities, so it does not require bagging per se. The inflated argmax on its own is very computationally efficient. Please see discussion point 2 in our global response for discussion on the computational cost of bagging.
>
>
> 3. *“Can we also see the performance of inflated argmax without bagging in the experiment? In my understanding, this should also be possible with the proposed efficient algorithm. This can be useful to see how important the bagging method is needed for the proposed algorithm.”*
>
>     We have added this comparison—please see the pdf attached to our global response. As we can see, both bagging and the inflated argmax are important for selection stability.
>
>
> 4. *“Visualizing the number of bags needed to perform reasonably in the experiment can also be useful. Because it can be expensive if we use more sophisticated algorithm or using larger dataset.”*
>
>     We have rerun the experiment with a larger number of bags—please see the pdf attached to our global response. As you can see, this dramatically improves selection stability, especially with the inflated argmax. In our revised appendix, we will investigate more thoroughly how the number of bags impacts stability in practice.
>
>
> 5. *“In deep learning, overconfident problem of probability can often be an issue when one wants to use probability score. How does this problem affect the proposed inflated argmax in practice?”*
>
>     This is an interesting question. Overconfident probability scores mean that the top label is ahead by a large margin, so the inflated argmax will frequently return a singleton if $\epsilon$ is sufficiently small. In our global response, per your suggestion, we have attached a pdf which shows the stability of the base algorithm with the inflated argmax. On its own, the inflated argmax does not lead to selection stability *precisely because of overconfident probabilities.* Bagging addresses the problem of overconfidence, and then the inflated argmax captures the fact that several labels have similar probability scores after bagging.
>
>
> 6. *“Can we know that it is likely that the second class to be chosen as a candidate class is likely to be the second-largest probability class?”*
>
>     Yes, this is guaranteed by the 3rd statement of Proposition 9.
>
>
> 7. *“Is it reasonable to say that precision is similar given different average precision and standard error provided (Lines 293--294)?”*
>
>     Yes, it is reasonable. For example, if we compare the average precision between argmax+subbagging and inflated-argmax+subbagging, the pooled Z-score is around 1.5, which is not statistically significant. Considering further that we’re making multiple comparisons in the table, it would be unreasonable to say the rates are very different.
>
>
> 8. *“I was wondering if there are other possible baselines in related work that could be used in the experiments?”* and *“Suggestions on simple baselines 7.1 Top-2 class prediction, size of beta will be 2. 7.2 Predicting classes until sum of probability becomes a certain constant, e.g., if we set threshold = 0.8 and prediction probability vector is [0.1, 0.1, 0.1, 0.2, 0.6], then we will pick the last two classes (0.2 and 0.6). This simple idea looks intuitive and gives flexible set and this is the most simple baseline I had in mind that could be effective when I first read this paper. How do you think about this baseline?”*
>
>     We have added both of your suggestions and others. Please see discussion point 1 in our global response.

---

> > ### Comment · Reviewer_LzW3 · 2024-08-12
> > **Thank you for the author feedback.**
> >
> > Thank you very much for the author's feedback. I have read the other reviews and the rebuttal. Thank you for the explanation and additional experiments.
> >
> > I agree with the Author's rebuttal 2 regarding the core contribution. The core contribution of this paper is laying the foundation for stability classification by introducing several notions on classifier stability mentioned in this paper. This idea is novel to the best of my knowledge. Bagging is indeed more expensive than without Bagging, but the paper did a great job demonstrating that a theoretically justified algorithm archives this notion of stability. Having a more efficient algorithm or more analysis of this problem setting can be further studied in the future.
> >
> > I appreciate that other algorithms that are not necessarily stable but somewhat reasonable to try in this problem setting were added in the experiment section. This can motivate the practical usefulness of inflated argmax.
> >
> > Classification with abstention/rejection is indeed another framework that has been well-studied to abstain from making a prediction when the classifier is not confident. I treat this research direction as a different direction for making classifiers more reliable by outputting a flexible-size prediction set with some guarantee. This approach can also help machine learning systems to collaborate with humans to have better judgment and therefore worth investigating further.
> >
> > For these reasons, I increase the score to 7 (Accept).

---

> > > ### Author Response · Authors · 2024-08-13
> > > **Thanks for your reply**
> > >
> > > We sincerely appreciate your constructive comments throughout this process. We look forward to integrating your suggestions into our revision.

---

### Official Review · Reviewer_m1xX · 2024-07-09

**Soundness:** 3
**Presentation:** 3
**Contribution:** 2
**Rating:** 7
**Confidence:** 4

**Summary:**

An algorithm is considered stable if small perturbations in the training data do not result in significant changes to its output. Stability has been previously explored in the context of regression. This paper extends this concept to the multiclass framework, where previous work has focused on stability in terms of output probability rather than label selection processes. Given that argmax is not a continuous function, it can lead to instability in the classifier. To address this issue, the authors first define stability for the multiclass domain and then demonstrate that a stable classifier alone is insufficient; a corresponding label selection strategy is also necessary. The authors propose a framework combining bagging and inflated argmax to derive such a combination, which they validate experimentally, showing that their approach achieves higher stability while maintaining precision at a minimal cost.

**Strengths:**

1. The authors introduce a novel definition of algorithmic stability in classification, shifting the focus from output probability to predicted labels. They demonstrate that, according to this definition, a stable classifier is insufficient for achieving overall stability. Furthermore, they establish a connection between this notion of stability and classical algorithmic stability concepts. Building on these insights, the authors propose an approach that leverages bagging and inflated argmax to achieve said stability.
2. It is essential to emphasize that the proposed framework operates without assuming any specific distribution on the data. Additionally, it does not rely on the number of classes or dimensionality of the covariates, making it versatile and applicable to various algorithms.
3.The experimental findings demonstrate the stability of the proposed framework. Moreover, they reveal the negative impact of argmax on stability. Notably, the results show that the proposed approach incurs only a minor penalty to precision, underscoring its effectiveness.

**Weaknesses:**

1.The authors point out my primary concern in their discussion section, highlighting the practicality of the proposed framework. Notably, bagging is an computationally expensive procedure, and while the authors suggest that parallelization can mitigate this issue, I agree that this does not necessarily translate to a reduction in overall cost. This becomes particularly challenging when dealing with large datasets commonly encountered in modern classifiers (e.g., image classification), where the sheer scale of computation required can be prohibitively expensive even with multiple GPUs.
2. The experimental section relies heavily on a single dataset and classifier combination. To fully demonstrate the benefits of the proposed framework, I believe it would be necessary to conduct a more comprehensive set of experiments. Specifically, I suggest that the authors provide additional evidence through experimentation to illustrate how the slight tradeoff in precision is justified by the gained stability in practice.

**Questions:**

1. From the definition of stability given in 4, it is evident that the authors trained the classifiers multiple times while removing training samples. How does the overall average of average precision look like in such cases? Such a metric might show the advantage the proposed framework.
2. My other main question is related to practicality. I would like to know the author's thought more regarding that limitation.

---

> ### Author Rebuttal · Authors · 2024-08-06
>
> 1. *“The authors point out my primary concern in their discussion section, highlighting the practicality of the proposed framework. Notably, bagging is an computationally expensive procedure, and while the authors suggest that parallelization can mitigate this issue, I agree that this does not necessarily translate to a reduction in overall cost. This becomes particularly challenging when dealing with large datasets commonly encountered in modern classifiers (e.g., image classification), where the sheer scale of computation required can be prohibitively expensive even with multiple GPUs.”* and *“My other main question is related to practicality. I would like to know the author's thought more regarding that limitation.”*
>
>     Please see discussion point 2 in our global response, and let us know if you have any additional concerns regarding practicality.
>
>
> 2. *“The experimental section relies heavily on a single dataset and classifier combination. To fully demonstrate the benefits of the proposed framework, I believe it would be necessary to conduct a more comprehensive set of experiments. Specifically, I suggest that the authors provide additional evidence through experimentation to illustrate how the slight tradeoff in precision is justified by the gained stability in practice.”*
>
>     We agree that additional experiments may be useful for a more detailed understanding of this tradeoff, and would be happy to add more experiments to the supplement to pose the same questions under different settings and different data generating processes, in our revision. However, we would like to point out that, in our view, this experimental comparison is not central to the main point of the paper. While from an empirical point of view, our method versus a competing method are simply two different points along the tradeoff between precision and stability, from the theoretical point of view they are not on equal footing at all: our method offers a theoretical guarantee of selection stability, and there is no guarantee of this type at all for competing options.
>
>
> 3. *“From the definition of stability given in 4, it is evident that the authors trained the classifiers multiple times while removing training samples. How does the overall average of average precision look like in such cases? Such a metric might show the advantage the proposed framework.”*
>
>     In the table below, we compare $\beta_{\text{prec}}$ (as defined in the paper) to the "overall average of average precision" over all $500$ leave-one-out models. In this experiment, there is not a substantial difference between $\beta_{\text{prec}}$ (the original precision measure) and $\beta^{\text{LOO}}_{\text{prec}}$ (this new measure).
>
> |  | Results for Base Algorithm $\mathcal{A}$ | | | Results with Subbagging $\widetilde{\mathcal{A}}_m$ | |
> | -------- | ------- | ------- | ------- | ------- | ------- |
> |          |  $\beta_{\text{prec}}$ | $\beta^{\text{LOO}}_{\text{prec}}$ | | $\beta_{\text{prec}}$ | $\beta^{\text{LOO}}_{\text{prec}}$ |
> |  $\text{argmax}$  |  0.879 (0.003)  |  0.884 (0.002)  | | 0.893 (0.003)  |  0.893 (0.003)  |
> |  $\text{argmax}^\varepsilon$  |  0.873 (0.003)  |  0.879 (0.003)  | | 0.886 (0.003)  |  0.886 (0.003)  |
> |  $\text{top-}2$  |  0.000 (0.000)  |  0.000 (0.000)  | | 0.000 (0.000)  |  0.000 (0.000)  |
> |  Thresholding $\Gamma^*_{0.8}$  |  0.756 (0.004)  |  0.761 (0.004)  | | 0.704 (0.005)  |  0.703 (0.005)  |
> |  NDC for $F_1$ [1]  |  0.832 (0.004)  |  0.836 (0.003)  | | 0.825 (0.004)  |  0.825 (0.004)  |

---

> > ### Comment · Reviewer_m1xX · 2024-08-13
> > **Response to author rebuttal**
> >
> > I thank the authors for taking the time to addressing my concerns. I also see that some of these concerns were also discussed with other reviewers. After going through all the discussions, I think that my concerns have been addressed quite a bit. Thus I have decided to improve my score.

---

### Official Review · Reviewer_sBU3 · 2024-07-10

**Soundness:** 3
**Presentation:** 2
**Contribution:** 2
**Rating:** 6
**Confidence:** 3

**Summary:**

This paper focuses on the stability of learning multiclass classfiers. It proposes a notion of selection stability for the learning algorithm as well as a modified argmax that returns a set of labels. Evaluations have been conducted using FashionMNIST and simple models.

**Strengths:**

- This paper focuses on the problem of classifier stability, which has high practical interest.
- This paper proposes a relaxed form of argmax, which contributes concretely to the problem.
- Propositions and theoretical relations to existing results are soundly presented.

**Weaknesses:**

- The presentation of the proposed contribution can be improved.
  - The core proposal of inflated argmax is introduced in section 3.2, which is too late.
  - Bagging is not part of proposal, and shows low relation to the contribution, which can be demonstrated using fewer space.
  - Additional properties of the inflated argmax is also not directly related to the problem.
  - "learning stability by leave-one-out data" and "prediction stability when using argmax for similar probabilities" seem to be mixed and not clearly addressed.
  - Some sentences such as `workflow is fundamentally incompatible with the goal of algorithmic stability` is too extreme, for example, simple streament using proper regularization terms can somehow mitigate the problem.
- The demonstration of empirical evaluation does not match the last sentence in abstract.
  - The combination of $\epsilon$-inflated argmax on the base learning algorithm should be compared.
  - It is also nice to show results on at least two datasets.

**Questions:**

- How tail stability and selection stability relates to $\epsilon$-compatibility differently? To confirm, do bagging classfiers only satisfy tail stablity? Are there other algorithm designs satisfy these stabilities?
- How to better interpreter the empirical evaluation results. With similar accuracy shown in Table 1, the proposed method should show better stability in Figure 2?

**Limitations:**

Practical limitations are presented in section 5.

---

> ### Author Rebuttal · Authors · 2024-08-06
>
> 1. *“The presentation of the proposed contribution can be improved.”*
>
>      a. *“The core proposal of inflated argmax is introduced in section 3.2, which is too late.”*
>
>       Thank you for this feedback. We agree that introducing inflated argmax earlier in the paper would be better and will try to reorganize if possible in order to do so. However, we would like to point out that the framework of selection stability and its connections to classical algorithmic stability are core contributions, and these are given in Section 2 -- in other words, the inflated argmax definition is not the first novel contribution presented in the paper.
>
>      b. *“Bagging is not part of proposal, and shows low relation to the contribution, which can be demonstrated using fewer space.”*
>
>      While the section on bagging is indeed background material, in order to be able to use bagging as the first stage of our two-stage procedure, we need to recap the requisite tail stability result for bagging -- this is an essential ingredient of our selection stability framework. However, in our revision we will add a sentence to the start of this section to clarify that this is background material and explain its role in what follows.
>
>      c. *“Additional properties of the inflated argmax is also not directly related to the problem.”*
>
>      Section 3.2.2 (on additional properties of the inflated argmax) covers a lot of highly relevant material. We show how to compute our inflated argmax. The properties in Proposition 9 illustrate how the inflated argmax in many ways acts as a natural extension of the argmax. Our optimality result, Proposition 11, connects back to these additional properties by establishing that the inflated argmax is the *most parsimonious $\epsilon$-compatible extension of the argmax*. In our revision, we will add clarifications to the text to explain the significance of these results and how they relate to the main aims of the paper.
>
>
>      d. *“‘learning stability by leave-one-out data’ and ‘prediction stability when using argmax for similar probabilities’ seem to be mixed and not clearly addressed.”*
>
>      Neither of these phrases are used in the paper. Could you please clarify what you mean?
>
>
>      e. *“Some sentences such as workflow is fundamentally incompatible with the goal of algorithmic stability is too extreme, for example, simple streament using proper regularization terms can somehow mitigate the problem.”*
>
>      It may be possible that regularization can make the predicted probabilities more stable. However, as we point out in Sections 1.1 and 2.3, this is not enough to ensure selection stability *because the argmax is discontinuous.* The issue is that *selection* is unstable whenever the model is nearly equally confident in multiple classes.
>
> 2. *“The demonstration of empirical evaluation does not match the last sentence in abstract.”*
>
>     The last sentence of the abstract states, "we demonstrate that the inflated argmax provides necessary protection against unstable classifiers, without loss of accuracy." This does match our empirical evaluation. In Figure 2 we see a dramatic improvement in stability (see also the revised Figure 2 attached as a pdf to the global rebuttal). While $\beta_{\text{prec}}$ in Table 1 decreases slightly with the inflated argmax, the difference is not statistically significant. Specifically, comparing the average precision between argmax+subbagging and inflated-argmax+subbagging, the pooled $Z$-score is around 1.5, which is not statistically significant. Despite having $N=10,000$ test samples, there is not a detectable difference in $\beta_{\text{prec}}$ before and after employing the inflated argmax. We will make this point more clearly in our revision.
>
>      a. *“The combination of 𝜖-inflated argmax on the base learning algorithm should be compared.”*
>
>      We have added this baseline to our table: please see discussion point 1 in our global response. We have also added it to the revised Figure 2, which is attached in our global response.
>
>
>      b. *“It is also nice to show results on at least two datasets.”*
>
>      We agree that additional experiments may be useful, and would be happy to add more experiments to the supplement to pose the same questions under different settings and different data generating processes, in our revision. However, the main contribution of our paper is theoretical: we show how to guarantee selection stability for *any classifier* on *any dataset*. Our emphasis is thus on the idea and the theoretical contribution. Our experiment in Section 4 and simulation in Section D.2 serve as illustrations of these results to show how they work in practice. That being said, we will add more experiments to the supplement to reinforce these results.
>
> 3. *“How tail stability and selection stability relates to 𝜖-compatibility differently? To confirm, do bagging classfiers only satisfy tail stablity? Are there other algorithm designs satisfy these stabilities?”*
>
>     For the first question, we address the relationship between all three criteria in Proposition 4. We can expand the discussion around Proposition 4 in our revision to explain the relationship more clearly. For the latter two questions, other meta-algorithms may satisfy tail stability. This is an interesting open question.
>
>
> 4. *“How to better interpreter the empirical evaluation results. With similar accuracy shown in Table 1, the proposed method should show better stability in Figure 2?”*
>
>     Our proposed method does show much better stability. Note in Figure 2 that smaller curves are more stable. In our revised Figure 2 (attached to our global response), the instability measure $\delta_j = 0$ for *every* test point for our proposed method.

---

> > ### Comment · Reviewer_sBU3 · 2024-08-11
> >
> > I thank authors detailed and patient response and I have confirmed my shared concern over limited experiments and computational costs of bagging.
> > I thank authors for clarifying the organization of the manuscript, as well as my missing attention on section 3.2.2 and proposition 4. I also thank the improved figure 2.
> > For my concern 1.e. for definition 1, I missed the notation of feature $x$ in $C$ and thought this considers only the classifiers sets output by an learning algorithm.

---

> ### Author Response · Authors · 2024-08-13
> **Thanks for your reply**
>
> We sincerely appreciate your constructive comments throughout this process. We look forward to integrating your suggestions into our revision.

---

### Official Review · Reviewer_FNaQ · 2024-07-10

**Soundness:** 3
**Presentation:** 3
**Contribution:** 3
**Rating:** 7
**Confidence:** 3

**Summary:**

This submission studies the problem of making set-valued predictions, in which the classifier should strive for an optimal balance between the correctness (the true class is among the candidates) and the precision (the candidates are not too many) of its prediction.

Yet, this submission mentions one method for this purpose [1], which predicts, for each query instance, a Bayes-optimal set valued prediction optimizing the F-measure. It seems to miss a discussion on (several) related methods, which are constructed based on both the probability theory and imprecise probability, e.g., the ones discussed in [3, 4, 6] and references therein.

On the abstract level, the inflated argmax presented in Section 3.2  might be seen as a way to distort/imprecise the given singleton conditional probability distribution. It might enlarge the set of existing approaches targeting similar purposes, such as the ones discussed in [2, 5] and references therein, meaningfully.

The proposed algorithm is compared with a variant of LeNet-5, implemented in PyTorch [PGML19] tutorials as GarmentClassifier(), and its ensemble version on the Fashion-MNIST data set. Empirical evidence suggests that the proposed algorithm provides better $\beta_{\text{prec}}$ than the single classifier, but worse than the ensemble version. The average set size produced by the proposed algorithm seems to be reasonably small.  Results on the selection instability $\delta_j$ defined in (4), i.e., an approximate of the loss version of the stability $\delta$ defined in (1), which essentially measures the robustness of the set-valued predictions under tiny changes in the training data set, seems to be in favor of the proposed algorithm.

[1] Del Coz, J. J., Díez, J., & Bahamonde, A. (2009). Learning Nondeterministic Classifiers. Journal of Machine Learning Research, 10(10).

[2] Montes, I., Miranda, E., & Destercke, S. (2020). Unifying neighbourhood and distortion models: part I–new results on old models. International Journal of General Systems, 49(6), 602-635.

[3] Mortier, T., Wydmuch, M., Dembczyński, K., Hüllermeier, E., & Waegeman, W. (2021). Efficient set-valued prediction in multi-class classification. Data Mining and Knowledge Discovery, 35(4), 1435-1469.

[4] Nguyen, V. L., Destercke, S., Masson, M. H., & Hüllermeier, E. (2018, July). Reliable multi-class classification based on pairwise epistemic and aleatoric uncertainty. In 27th International Joint Conference on Artificial Intelligence (IJCAI 2018) (pp. 5089-5095).

[5] Nguyen, V. L., Zhang, H., & Destercke, S. (2023, September). Learning Sets of Probabilities Through Ensemble Methods. In European Conference on Symbolic and Quantitative Approaches with Uncertainty (pp. 270-283). Cham: Springer Nature Switzerland.

[6] Troffaes, M. C. (2007). Decision making under uncertainty using imprecise probabilities. International journal of approximate reasoning, 45(1), 17-29.

**Strengths:**

S1: I think taking into account the instability, due to tiny changes in the training data set, when making set-valued prediction is interesting. Yet, assessing this aspect of classifiers seems to be costly, i.e., requiring a sufficiently large number of leave-one-out models.

S2: Empirical evidence seems to be promising. For example, $\beta_{\text{prec}}$ seems to suggest that the proposed algorithm tends to produce a reasonably high proportion of correct singleton predictions. The average set size seems to suggest that the set-valued predictions produced by the proposed algorithms are reasonably small. Empirical evidence regarding the instability defined in (4) seems to be in favor of the proposed algorithm.

**Weaknesses:**

W1: I think adding a few more competitors would be useful in assessing the potential advantages of the proposed algorithms. For example, threshold-based algorithms, which produce Bayes-optimal predictions of F-measure [1], and $U_{65}$ and $U_{80}$ [3], would be a good choice. They would cost $O(L\log(L))$ time to sort the labels according to the decreasing order of the predicted conditional probability masses and select the optimal threshold.

W2: Assessing the classifiers with respect to utility-discounted predictive accuracies, such as $U_{65}$ and $U_{80}$ which respectively give small and large rewards for being cautious [7] , would help to further highlight the potential (dis)advantages of the classifiers.

W3: Moreover, an imprecise classifier should abstain (i.e., provide set-valued predictions) on difficult cases, on which the precise classifier is likely to fail. This would be assessed in different ways. For example, one might consider reporting the correctness of the cautious classifiers in the case of abstention versus the accuracy of the precise classifiers [4].

W4: The tolerance $\epsilon$ is set to be $.05$. Please provide arguments supporting this choice. Yet, choosing this hyperparameter using a validation set might be challenging. Providing a sensitivity analysis on the choice of  $\epsilon$ would be helpful.

[1] Del Coz, J. J., Díez, J., & Bahamonde, A. (2009). Learning Nondeterministic Classifiers. Journal of Machine Learning Research, 10(10).

[3] Mortier, T., Wydmuch, M., Dembczyński, K., Hüllermeier, E., & Waegeman, W. (2021). Efficient set-valued prediction in multi-class classification. Data Mining and Knowledge Discovery, 35(4), 1435-1469.

[4] Nguyen, V. L., Destercke, S., Masson, M. H., & Hüllermeier, E. (2018, July). Reliable multi-class classification based on pairwise epistemic and aleatoric uncertainty. In 27th International Joint Conference on Artificial Intelligence (IJCAI 2018) (pp. 5089-5095).

[7] Zaffalon, M., Corani, G., & Mauá, D. (2012). Evaluating credal classifiers by utility-discounted predictive accuracy. International Journal of Approximate Reasoning, 53(8), 1282-1301.

**Questions:**

Q1: Please refer to "Weaknesses" for detailed comments and suggestions for further revision.

Q2: The notion of selection stability $\delta$ at sample size $n$ seems to be interesting. I guess the experimental setting provides the necessary information to compute the value of the selection stability $\delta$ given in (1) . To provide an idea of how empirical evidence may differ from the theoretical results, it might be helpful to compute that value and report the proportion of test instances whose empirical values $1 - \delta_j$ are smaller than $\delta$.

Q3: I guess re-defining (4) as $\delta_j = \frac{1}{500}\sum_{k=1}^{500} 1{s(\hat{p}(\tilde{X}_j)) \cap s(\hat{p}^{\setminus i_k}(\tilde{X}_j)) = \emptyset}$ and changing the description of Figure 2 accordingly (to be consistent with the notion of the selection stability $\delta$ defined in Definition 1) might make things a bit clearer.

**Limitations:**

L1: Discussions on related work might need to be extended.

L2:  the empirical study might need to be enlarged with closely related algorithms.

Please refer to "Weaknesses" for detailed comments and suggestions for further revision.

---

> ### Author Rebuttal · Authors · 2024-08-05
>
> 1. *“W1: I think adding a few more competitors would be useful in assessing the potential advantages of the proposed algorithms…”*, *“W2: Assessing the classifiers with respect to utility-discounted predictive accuracies, …”* and “L2: the empirical study might need to be enlarged with closely related algorithms.”
>
>     Please see discussion point 1 in our global response. Below we compare algorithms based on $u_{65}$ and $u_{80}$ per your suggestion in W2. However, we stress that the inflated argmax $\text{argmax}^\varepsilon$ is *not* meant to compete with any of these alternative set-valued classification methods, nor is it optimizing for $u_{65}$ or $u_{80}$. It is meant to improve upon the argmax by guaranteeing selection stability. None of these other methods have been shown to satisfy selection stability, and even if they were, ours satisfies the optimality result of Proposition 11.
>
> || Results for |Base Algorithm $\mathcal{A}$||| Results with |Subbagging $\widetilde{\mathcal{A}}_m$||
> | --- | --- | --- | --- | --- | --- | --- |  --- |
> | |  $u_{65}$ | $u_{80}$ | $\beta_{\text{size}}$ | | $u_{65}$ | $u_{80}$ | $\beta_{\text{size}}$ |
> | $\text{argmax}$  |  0.879 (0.003)  |  0.879 (0.003)  |  1.000 (0.000)  | | 0.893 (0.003)  |  0.893 (0.003)  | 1.000 (0.000)  |
> | $\text{argmax}^\varepsilon$  |  0.881 (0.003)  |  0.883 (0.003)  |  1.015 (0.001)  | | 0.895 (0.003)  |  0.897 (0.003)  | 1.015 (0.001)  |
> | $\text{top-}2$  |  0.632 (0.001)  |  0.777 (0.001)  |  2.000 (0.000)  | | 0.631 (0.001)  |  0.777 (0.001)  | 2.000 (0.000)  |
> | Thresholding $\Gamma^*_{0.8}$  |  0.880 (0.002)  |  0.909 (0.002)  |  1.248 (0.005)  | | 0.868 (0.002)  |  0.907 (0.002)  | 1.248 (0.005)  |
> | NDC for $F_1$ [1] | 0.889 (0.003)  |  0.902 (0.003)  |  1.105 (0.003)  | | 0.898 (0.003)  |  0.915 (0.002)  | 1.105 (0.003)  |
>
> 2. *“W3: Moreover, an imprecise classifier should abstain (i.e., provide set-valued predictions) on difficult cases, on which the precise classifier is likely to fail. This would be assessed in different ways. For example, one might consider reporting the correctness of the cautious classifiers in the case of abstention versus the accuracy of the precise classifiers [4].”*
>
>     The idea of "abstention" is a useful framework, but in fact, this does actually appear implicitly for any a set valued classifier — if we return the entire set $\hat{S} = $ {$1,\dots,L$}, this means that we are not able to say anything about the true value of the label, i.e., this is equivalent to an abstention. In fact, set-valued classification has the capacity to return strictly more information than a label-or-abstain scheme: if we always either return a label (i.e., $\hat{S} =$ {$y$} for some $y$) or an abstention (equivalently, $\hat{S} =$ {$1,\dots,L$}), we do not have the opportunity to express partial information (for example, $\hat{S} =$ {$y_1,y_2$} — this allows us to express that we have partial information about the true label for a particular sample $X$, i.e., we are allowing adaptivity to our level of uncertainty in different instances).
>
>     Turning to the question of empirical comparison, in a label-or-abstain scheme, measuring the frequency of abstaining is equivalent to measuring how often $|\hat{S}|=1$, which is related to our precision measure in our experiments.
>
> 3. *“W4: The tolerance $\epsilon$ is set to be .05. Please provide arguments supporting this choice. Yet, choosing this hyperparameter using a validation set might be challenging. Providing a sensitivity analysis on the choice of $\epsilon$ would be helpful.”*
>
>     The tolerance parameter $\epsilon$ has an intrinsic meaning; it is different from a tuning parameter such as the regularization parameter $\lambda$ for Lasso where there is no intrinsic meaning and therefore tuning is the only reasonable way to choose a value. In our setting, $\epsilon$ represents the margin by which we inflate the argmax — for example setting $\epsilon$ = 0.05 can, in a sense, be viewed as saying that an estimated probability of 0.45 versus 0.4 is sufficiently close to be ambiguous in terms of which label should be the “winner”. For this reason, it does not seem appropriate to perform a sensitivity analysis.  (Of course, as we see in our theory, it’s also possible for the user to instead choose a desired value of $\delta$, and then specify $\epsilon$ accordingly via Theorem 5; again, here $\epsilon$ would then be determined by the user’s desired value of $\delta$, and does not need to be tuned.)
>
> 4. *“Q2: The notion of selection stability $\delta$ at sample size $n$ seems to be interesting. I guess the experimental setting provides the necessary information to compute the value of the selection stability $\delta$ given in (1) . To provide an idea of how empirical evidence may differ from the theoretical results, it might be helpful to compute that value and report the proportion of test instances whose empirical values... are smaller than $\delta$.”*
>
>     Note that various $\delta_j$ range over different points in the test set $X_j$, and selection stability holds for all test points $x$. Thus, selection stability means that *all* of the $\delta_j$ are at most $\delta$. We will clarify this point in our discussion of the experiment results in our revision.
>
> 5. *“Q3: I guess re-defining (4) as... and changing the description of Figure 2 accordingly (to be consistent with the notion of the selection stability $\delta$ defined in Definition 1) might make things a bit clearer.”*
>
>     Thank you for catching this—we will fix the definition in our revision.
>
> 6. *“L1: Discussions on related work might need to be extended.”*
>
>     In discussion point 1 of our global response, we discuss other set-valued classification methods that we have added to the experiments. We will add discussion of these methods to the related work section, including all of the references you suggest. We also welcome any additional suggestions for related work that we should discuss.

---

> ### Comment · Reviewer_FNaQ · 2024-08-09
>
> Thanks for your detailed response. Additional empirical evidence and discussions indeed suggest more about the potential (dis)advantages of the proposed algorithms.
>
> Based on the response, I think we might see  $\beta_{\text{pre}}$ as an evaluation metric for classification with rejection, i.e., a predictor will only make a prediction for a query instance if it can make a precise/singleton prediction. This might also imply that whether the set-valued prediction covers the true class or not and whether the set size is large or not are not relevant. In this sense, I think it is fair to say that the proposed algorithms provide advantages, compared to top-2, Thresholding $\Gamma^*_{80}$ and NDC for $F_1$. From their definitions, it would be not hard to see that the set-valued predictions that optimize $u_{65}$, $F_1$, $u_{80}$ and a few other criteria, e.g., ones mentioned in [6], cover the top-ranked class, i.e., the output of $argmax$ (unless there are multiple classes with the highest score). By definition, optimizing $u_{65}$ would provide smaller set-valued predictions, compared to others, and therefore might naturally become a related competitor regarding $\beta_{\text{pre}}$.
>
> Given the (set-valued) predictions on the test set, the test set can be partitioned into 2 parts, where the classifier/predictor makes precise predictions and set-valued predictions.  As far as I understand from [7] (and [3]), $u_{65}$, $u_{80}$, and other utility-discounted predictive accuracies are designed to assess classifiers/predictors on the entire (test) data set. The additional results suggest that the proposed algorithms provide worse scores compared to NDC for $F_1$, which is also not designed to optimize both $u_{65}$ and $u_{80}$. If one opts for $u_{65}$ and $u_{80}$, I think it is reasonable to optimize them directly. Yet, I think it is difficult to find some specific evaluation criterion/criteria to assess multiple aspects of set-valued predictors. I think it is reasonable to expect a proposed set-valued predictor to demonstrate the advantages regarding at least a few criteria that assess the predictor on the entire (test) data set.
>
> Regarding "W3: Moreover, an imprecise classifier should abstain (i.e., provide set-valued predictions) on difficult cases, on which the precise classifier is likely to fail. This would be assessed in different ways. For example, one might consider reporting the correctness of the cautious classifiers in the case of abstention versus the accuracy of the precise classifiers [4].", I guess I should make the point more explicit. As far as I understand, the correctness of the cautious classifiers is the proportion of the time the set-valued predictions cover the true classes. I think such a criterion is designed to assess the cost of making set-valued predictions, i.e., the cautious classifiers should (only) provide set-valued predictions on difficult cases, on which the precise classifier is likely to fail. In my opinion, this criterion should complement the $\beta_{\text{pre}}$, utility-discounted predictive accuracies, set size, and other evaluation criteria, which take into account both precise predictions and set-valued predictions.
>
> Yet, I think the notion of selection stability is interesting (as I wrote in the initial review), and it might make the proposed algorithms differ from other cautious classifiers or set-valued predictors. The motivation to go with the proposed algorithms might need to be further elaborated. After due consideration, I keep my initial rating, but I appreciate any additional results and discussions that may strengthen the motivation to go with the proposed algorithms.

---

> > ### Author Response · Authors · 2024-08-13
> >
> > Thank you for your further detailed comments. We have expanded our table to include the two methods you suggest, Set-Valued Bayes-Optimal Prediction (SVBOP) for $u_{65}$ and $u_{80}$. We bold the statistical winners in each column.$^\dagger$ These new methods overall perform somewhat similar to NDC for $F_1$, but as you anticipated, the expected set size for SVBOP-65 is the smallest of the three. However, the expected set size for the inflated argmax is much smaller than these other three methods. Note also that although its $u_{65}$ utility *is* slightly smaller than SVBOP-65 on the test set, the inflated argmax qualifies as a statistical winner in this column as well. Our proposal thus performs competitively along three of the four criteria we have considered ($\beta_{\text{prec}}$, $\beta_{\text{size}}$ and $u_{65}$).
> >
> > We agree that, on its own, we can view $\beta_{\text{prec}}$ as an evaluation metric for classification with rejection, since it requires a singleton prediction. Our motivating goal is to return a singleton as often as possible while achieving a user-specified selection stability level. While there are many different perspectives on set-valued classification, we reiterate that what is most novel about our work is that our proposal has distribution-free theory. This means that, regardless of the dataset or base algorithm used, we can guarantee that our method will be stable. Furthermore, our optimality result shows that the inflated argmax is singleton as often as possible among all $\epsilon$-compatible selection rules.
> >
> > | | Results for Base Algorithm $\mathcal{A}$| | | | | Results with Subbagging $\widetilde{\mathcal{A}}_m$| | | |
> > | --- | --- | --- | --- | --- | --- | --- | --- | --- | --- |
> > |          |  $u_{65}$ | $u_{80}$ | $\beta_{\text{prec}}$ | $\beta_{\text{size}}$ | | $u_{65}$ | $u_{80}$ | $\beta_{\text{prec}}$ | $\beta_{\text{size}}$ |
> > |  $\text{argmax}$  |  0.879 (0.003)  |  0.879 (0.003)  |  **0.879** (0.003) | **1.000** (0.000)  | | **0.893** (0.003)  |  0.893 (0.003)  | **0.893** (0.003)  |  **1.000** (0.000)  |
> > |  $\text{argmax}^\varepsilon$  |  **0.881** (0.003)  |  0.883 (0.003)  |  **0.873** (0.003)  |  1.015 (0.001)  | | **0.895** (0.003)  |  0.897 (0.003)  | **0.886** (0.003)  |  1.017 (0.001)  |
> > |  $\text{top-}2$  |  0.632 (0.001)  |  0.777 (0.001)  |  0.000 (0.000)  |  2.000 (0.000)  | | 0.631 (0.001)  |  0.777 (0.001)  | 0.000 (0.000)  |  2.000 (0.000)  |
> > |  Thresholding $\Gamma^*_{0.8}$  |  0.880 (0.002)  |  **0.909** (0.002)  |  0.756 (0.004)  |  1.248 (0.005)  | | 0.868 (0.002)  |  0.907 (0.002)  | 0.703 (0.005) | 1.357 (0.006)  |
> > |  NDC for $F_1$  |  **0.889** (0.003)  |  **0.902** (0.003)  |  0.832 (0.004)  |  1.105 (0.003)  | | **0.898** (0.003)  |  **0.915** (0.002)  | 0.825 (0.004) | 1.137 (0.004)  |
> > |  SVBOP for $u_{65}$  |  **0.889** (0.003)  |  0.901 (0.003)  |  0.838 (0.004) | 1.091 (0.003)  | | **0.900** (0.003)  |  **0.915** (0.002)  | 0.835 (0.004) | 1.117 (0.003)  |
> > |  SVBOP for $u_{80}$  |  **0.885** (0.003)  |  **0.910** (0.002)  |  0.777 (0.004) | 1.190 (0.004)  | | 0.882 (0.002)  |  **0.915** (0.002)  | 0.744 (0.004) | 1.245 (0.005)  |
> >
> > Regarding your second suggestion, we looked at, for each imprecise classifier $\hat{s}$, the ratio of "accuracy of standard argmax given $\hat{s}$ abstains" divided by "accuracy of $\hat{s}$ given $\hat{s}$ abstains". We call this the *superfluous inflation*, since if this ratio is close to 1, it means the argmax is correct as often as the set-valued classifier (when the latter abstains), so outputting a set could be seen as overly conservative. We include the table with these results below. The inflated argmax has the smallest ratio, meaning it only abstains in hard cases where it can improve the accuracy by returning a non-singleton set.
> >
> > | | Results for Base Algorithm $\mathcal{A}$ | Results with Subbagging $\widetilde{\mathcal{A}}_m$ |
> > | --- | --- | --- |
> > | |  $\beta_{\text{superfluous-inflation}}$ | $\beta_{\text{superfluous-inflation}}$ |
> > |  $\text{argmax}^\varepsilon$  |  **0.496** (0.005)  |  **0.504** (0.005)  |
> > |  $\text{top-}2$ | 0.905 (0.003) | 0.919 (0.003) |
> > |  Thresholding $\Gamma^*_{0.8}$  |  0.622 (0.005) |  0.702 (0.005)  |
> > |  NDC for $F_1$ | 0.531 (0.005)  |  0.591 (0.005)  |
> > |  SVBOP for $u_{65}$ | 0.525 (0.005)  |  0.575 (0.005)  |
> > |  SVBOP for $u_{80}$ | 0.611 (0.005)  |  0.689 (0.005)  |
> >
> >
> >
> > $^\dagger$ To check whether a given method wins along a given column, we compute the two-sample Z-score, subtracting that method’s value from the highest value in that column and normalizing by the pooled standard error. We say method is a winner if the two-tailed Z-test is *not* statistically significant at 0.05.

---

> > > ### Comment · Reviewer_FNaQ · 2024-08-13
> > >
> > > Thank you for the additional empirical evidence and discussions. To accommodate these, I increase the rating to $7$.

---

> > > > ### Author Response · Authors · 2024-08-13
> > > > **Thanks for your replies**
> > > >
> > > > We sincerely appreciate your constructive comments throughout this process. We look forward to integrating your suggestions into our revision.

---

### Official Review · Reviewer_mVCr · 2024-07-13

**Soundness:** 2
**Presentation:** 2
**Contribution:** 2
**Rating:** 4
**Confidence:** 3

**Summary:**

The paper considers the stability of classifiers in multi-class classification. In the considered framework, the classifier is allowed to return a set of candidate classes instead of only one class. The stability is defined as the frequency of a set predicted by one classifier having no union with a set predicted with a classifier trained on the same training set with only one sample removed. The authors proposed the construction of a stable classifier in this framework using bagging and inflated argmax. The proposed approach is proven to have stability guarantees. Additionally, the effectiveness of the approach in terms of increased stability is confirmed in the empirical experiment.

**Strengths:**

- The paper is nicely written and easy to follow.
- The inflated argmax is shown theoretically to be the maximizer of classifier stability.

**Weaknesses:**

- The authors argue that the argmax is a hard operator that may make the classifier unstable if even small changes of the outputs occur, but the definition of stability used in the paper also uses the hard operator of the set union being equal to an empty set.
- The definition of the stable classifier as defined in this paper is new to me, and based on the paper, I struggle to find a good motivation for using this specific definition and focusing on it.
- The method requires bagging, what makes it difficult to apply with heavy architectures.
- The empirical results are limited to a single experiment.
- The empirical experiment uses baselines that do not work in the framework of set-valued prediction and always predict only one class, which puts them at a clear disadvantage under the considered task. The comparison should include other set-valued classifiers. E.g., classifiers mentioned in related works or classifiers that optimize simple set-utility functions like in [1] and [2], that can be very efficiently optimized as discussed in [3].

[1] Del Coz JJ, Díez J, Bahamonde A. Learning nondeterministic classifiers. 2009
[2] Zaffalon M, Giorgio C, Maua DD. Evaluating credal classifiers by utility-discounted predictive accuracy. 2012
[3] Mortier T, Wydmuch M, Dembczynski K, Hullermeier E, Waegeman W. Efficient set-valued prediction in multi-class classification. 2021


NITs:
- "The more recent literature has focused on producing a sparse set of weights, but none of these works offer a formal stability guarantee for the support of the weights." This statement needs citation.

**Questions:**

- What are the specific applications benefiting from this notation of stability?
- How does the introduced inflated argmax compare with other set-valued classifiers?

**Limitations:**

Limitations are discussed. I see no negative social impact of this work.

---

> ### Author Rebuttal · Authors · 2024-08-05
>
> 1. *“The authors argue that the argmax is a hard operator… but the definition of stability used in the paper also uses the hard operator of the set union being equal to an empty set”*
>
>     The argmax does not satisfy $\epsilon$ compatibility. The point of the paper is to introduce an $\epsilon$ compatible relaxation of the argmax, which can be combined with bagging to stabilize any classifier. This relaxation is necessary even though the definition of stability uses a hard operator.
>
> 2. *“The definition of the stable classifier … I struggle to find a good motivation for using this specific definition and focusing on it.”* and *“What are the specific applications benefiting from this notation of stability?”*
>
>     Selection stability controls the probability that our algorithm makes contradictory claims when dropping a single data point at random from the training set. In set-valued classification, $\hat{S}$ represents the set of candidate labels returned by the model. This set should not totally change just by dropping a single observation from the training set. In our revision, we will expand Section 2 of the paper with a more extensive discussion motivating this definition.
>
> 3. *“The method requires bagging, what makes it difficult to apply with heavy architectures.”*
>
>     Please see discussion point 2 in our global response.
>
>
> 4. *“The empirical results are limited to a single experiment.”*
>
>     The main contribution of our paper is theoretical: we show how to guarantee selection stability for any classifier. Our emphasis is thus on the idea and the theoretical contribution. Our experiment in Section 4 and simulation in Section D.2 serve merely as illustrations of these results to show how they work in practice.
>
>
> 5. *“The empirical experiment uses baselines that do not work in the framework of set-valued prediction…”* and *“How does the introduced inflated argmax compare with other set-valued classifiers?”*
>
>     Please see discussion point 1 in our global response.
>
> 6. *“‘The more recent literature…’ This statement needs citation.”*
>
>     We should have said ‘the more recent papers in this line of work’ to make it clear that we were referencing the specific citations in the previous sentence. We will make this clearer in the revision.

---

### Author Rebuttal · Authors · 2024-08-06

We thank all of the reviewers for their insightful comments and effort in reviewing the paper. Below we discuss two topics that came up in multiple reviews.

1. **Improvements to the experiments section.** Based on feedback from several reviewers, we have expanded our experiments section by adding some additional existing methods as baselines to our table of results. We added several methods for set-valued classification suggested by the reviewers.

    The old version of our experiment served to illustrate what would happen if, as is most common in classification, you only return a single label. The experiment showed that practical learning algorithms can indeed be unstable for selection, even with bagging. The experiment illustrates our theory that the inflated argmax, combined with bagging, can ameliorate this issue for any dataset and any learning algorithm.

    Based on feedback, we added the following baselines:
     - Top-2 class prediction.
     - Thresholding: Predicting classes until sum of probability becomes at least 0.8,
     - Non-deterministic classification (NDC) [1] optimized for $F_1$-score.

    We stress, however, that these methods are not in direct competition with the inflated argmax since, since our stability framework is one of the main contributions of our paper. That is, no existing method for set-valued classification can guarantee selection stability.

    Below we present the updated table of results. Recall that $\beta_{\text{prec}}$ is the precision, meaning the probability that the algorithm returns the correct singleton, and $\beta_{\text{size}}$ is the average size of the set of candidate labels. Note that the inflated argmax is smaller than all competing set-valued methods and has a higher value of $\beta_{\text{prec}}$.

    We also reran the experiment with a larger number of bags—**please see the attached pdf for the stability curves**. We find that increasing the number of bags improves stability even further, especially with the inflated argmax.

|  | Results for Base Algorithm $\mathcal{A}$ || | Results with Subbagging $\widetilde{\mathcal{A}}_m$ |  |
| -------- | ------- | ------- | ------- | ------- | ------- |
|          |  $\beta_{\text{prec}}$ | $\beta_{\text{size}}$ | | $\beta_{\text{prec}}$ | $\beta_{\text{size}}$ |
|  $\text{argmax}$  |  0.879 (0.003)  |  1.000 (0.000)  | | 0.893 (0.003)  |  1.000 (0.000)  |
|  $\text{argmax}^\varepsilon$  |  0.873 (0.003)  |  1.015 (0.001)  | | 0.886 (0.003)  |  1.016 (0.001)  |
|  $\text{top-}2$  |  0.000 (0.000)  |  2.000 (0.000)  | | 0.000 (0.000)  |  2.000 (0.000)  |
|  Thresholding $\Gamma^*_{0.8}$  |  0.756 (0.004)  |  1.248 (0.005)  | | 0.704 (0.005)  |  1.354 (0.006)  |
|  NDC for $F_1$ [1]  |  0.832 (0.004)  |  1.105 (0.003)  | | 0.825 (0.004)  |  1.137 (0.004)  |

2. **Computational cost of bagging.**   Several reviewers pointed out that bagging may be computationally very demanding for modern large-scale algorithms. We would like to discuss this point here in a bit more detail (and will also add discussion into our revised paper).

    First, our paper is the first work demonstrating that guaranteed black-box classifier stability is possible at all — certainly it is very interesting and important to consider whether there are computationally efficient methods that can achieve this aim, but a first question is whether the aim is achievable at all. Our paper lays out the key definitions and theoretical groundwork for this problem, opening up new avenues for future research.  Moreover, if more efficient learning algorithms could in future be shown to satisfy tail stability (i.e., without bagging) then the main contribution of our paper—the inflated argmax—will be equally applicable and relevant. The inflated argmax is computationally efficient and, due to the two-stage structure of our theory, can be combined with any method shown to have tail stability at the first stage, i.e., can be separated from bagging.

    Next, to return to the question of bagging, we would like to justify why we feel that using bagging for the first stage of the procedure may in many cases be very feasible from the computational point of view. While the original definition of bagging uses the conventional bootstrap, where each bag contains as many samples as the original data set, i.e. $m = n$, in our framework we allow for arbitrary bag size $m$, which could be much smaller than the sample size $n$. Massively subsampling the data ($m \ll n$) can actually help scale learning algorithms to large data sets [2]. Moreover, bagging with $m \approx n$ can be expensive, but there are still many areas of machine learning where it is used, notably in Random Forests. Finally, our experiments also show that a modest number of bags ($B=200$) is all that we really need to start seeing major gains in selection stability.



References:

[1] Juan José Del Coz, Jorge Díez, and Antonio Bahamonde (2009). "Learning Nondeterministic Classifiers." *Journal of Machine Learning Research* 10(10).

[2] Ariel Kleiner, Ameet Talwalkar, Purnamrita Sarkar, and Michael I. Jordan (2014). "A scalable bootstrap for massive data." *Journal of the Royal Statistical Society Series B: Statistical Methodology* 76(4): 795-816.

---

### Decision · Program_Chairs · 2024-09-25

**Decision:**

Accept (poster)

**Comment:**

The reviewers have essentially converged around the idea that the paper is a worthy addition to NeurIPS; the paper indeed contains an interesting theoretical framework to build stability, and I commend the authors for elaborating further on experiments at rebuttal / discussion time.

At camera-ready time, I strongly suggest that the authors integrate not just their additional experimental results, but also make clear(er) point 2. in their rebuttal.